# No Metric to Rule Them All:
# Toward Principled Evaluations of Graph-Learning Datasets

Corinna Coupette [* 1 2]   Jeremy Wayland [* 3 4]   Emily Simons [3 4]   Bastian Rieck [3 4 5]

## Abstract

Benchmark datasets have proved pivotal to the success of graph learning, and *good* benchmark datasets are crucial to guide the development of the field. Recent research has highlighted problems with graph-learning datasets and benchmarking practices—revealing, for example, that methods which ignore the graph structure can outperform graph-based approaches. Such findings raise two questions: (1) What makes a good graph-learning dataset, and (2) how can we evaluate dataset quality in graph learning? Our work addresses these questions. As the classic evaluation setup uses datasets to evaluate models, it does not apply to dataset evaluation. Hence, we start from first principles. Observing that graph-learning datasets uniquely combine two modes—graph structure and node features—, we introduce RINGS, a flexible and extensible *mode-perturbation framework* to assess the quality of graph-learning datasets based on *dataset ablations*—i.e., quantifying differences between the original dataset and its perturbed representations. Within this framework, we propose two measures—*performance separability* and *mode complementarity*—as evaluation tools, each assessing the capacity of a graph dataset to benchmark the power and efficacy of graph-learning methods from a distinct angle. We demonstrate the utility of our framework for dataset evaluation via extensive experiments on graph-level tasks and derive actionable recommendations for improving the evaluation of graph-learning methods. Our work opens new research directions in data-centric graph learning, and it constitutes a step toward the systematic *evaluation of evaluations*.

*Equal contribution. [1]Aalto University, Finland [2]Max Planck Institute for Informatics, Germany [3]Helmholtz Munich, Germany [4]TU Munich, Germany [5]University of Fribourg, Switzerland. Correspondence to: Corinna Coupette <corinna.coupette@aalto.fi>.

*Proceedings of the 42$^{nd}$ International Conference on Machine Learning*, Vancouver, Canada. PMLR 267, 2025. Copyright 2025 by the author(s).

## 1. Introduction

Over the past decade, graph learning has established itself as a prominent approach to making predictions from relational data, with remarkable success in areas from small molecules (Fang et al., 2022; Stokes et al., 2020) to large social networks (Sharma et al., 2024; Ying et al., 2018). Despite significant progress on the theory of graph neural networks (Morris et al., 2024), however, many empirical intricacies of graph-learning tasks, models, and datasets remain poorly understood. For example, recent research has revealed that (1) purported performance gaps disappear with proper hyperparameter tuning (Tönshoff et al., 2023), (2) popular graph-learning datasets occupy a very peculiar part of the space of all possible graphs (Palowitch et al., 2022), (3) some graph-learning tasks can be solved without using the graph structure (Errica et al., 2020), and (4) graph-learning models struggle to ignore the graph structure when the features alone are sufficiently informative for the task at hand (Bechler-Speicher et al., 2024). These findings suggest a need for better infrastructure to assess graph-learning methods, supporting rigorous evaluations that paint a realistic picture of the progress made by the community.

**Necessity and Challenges of Dataset Evaluation.** Benchmark datasets play a key role in the evaluation of graph-learning methods, but the results cited above highlight that not all (collections of) graphs are equally suitable for that purpose. This motivates us to *flip the script* on graph-learning evaluation, asking how well graph-learning datasets can characterize the capabilities of graph-learning methods, rather than how well these methods can solve tasks on graph-learning datasets. Our work is guided by two questions:

**Q1** What characterizes a good graph-learning dataset?
**Q2** How can we evaluate dataset quality in graph learning?

Addressing these questions is not straightforward. First, the classic evaluation setup, which compares performance across models while *holding the dataset constant*, cannot be used to evaluate datasets. Second, comparing performance levels across datasets while *holding the model constant* yields measurements that are confounded by model capabilities. Third, while performance levels indicate the *difficulty* of a dataset for existing methods, these levels provide little

information about dataset *quality*: A difficult dataset of high quality may guide the field toward methodological innovation, but a difficult dataset of low quality may detract from real progress. Hence, our work starts from first principles.

**Desirable Properties of Graph-Learning Datasets.** We observe that attributed graphs combine two types of information, the *graph structure* and the *node features*. Graph-learning methods leverage both of these *modes* to tackle a given learning task.[1] This suggests the following *desirable property* for a dataset to reveal powerful insights into the capabilities of graph-learning methods, given a specific task:

**P0** The graph structure and the node features should contain *complementary task-relevant* information.

Assessing whether this property is present poses theoretical and practical challenges—not only due to the limitations of existing graph-learning methods but also because the relationship between task relevance and complementarity is potentially complicated. However, we can identify the following *necessary conditions* for **P0** to be satisfied:

**P1** The graph structure and the node features should both contain *task-relevant* information.
**P2** The graph structure and the node features should contain *complementary* information.

Notably, while **P1** is *task-dependent*, **P2** is *task-independent*.

**Principled Evaluations via Mode Perturbations.** Both **P1** and **P2** address the *relationship between the different modes* of a graph-learning dataset. Therefore, to test datasets for these properties, we propose RINGS (Relevant Information in Node features and Graph Structure), a dataset-evaluation framework based on the concept of *mode perturbation*. As illustrated in Figure 1, a mode perturbation maps an attributed graph $(G, X)$ to an attributed graph $(G', X')$, replacing the original edge set or feature set with a modified version according to a given transformation (e.g., randomization). This allows us to make measurements on both $(G, X)$ and $(G', X')$. Given appropriate measures, the difference between the resulting measurements can then provide insights into **P1** and **P2**. In analogy to model ablations in the evaluation of graph-learning methods, mode perturbations can also be thought of as *dataset ablations*.

**Our Contributions.** We make four main contributions:

**C1** **Framework.** We develop RINGS, a flexible and extensible framework to assess the quality of graph-learning datasets by quantifying differences between the original dataset and its perturbed representations.
**C2** **Measures.** As part of RINGS, we introduce two measures for evaluating graph datasets based on mode per-

---

[1]Notably, to be amenable to graph learning, even non-attributed graphs need to be assigned node features (e.g., one-hot encodings).

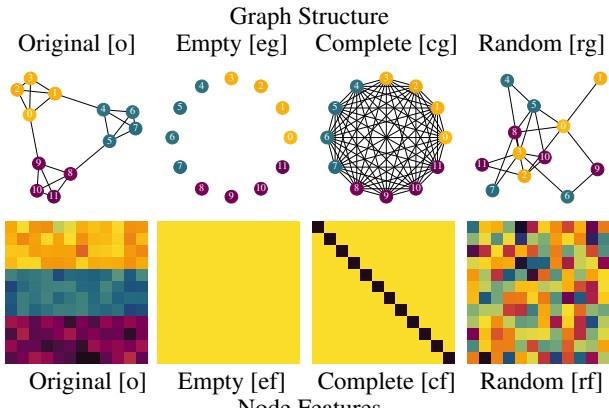

Figure 1: **Overview of our main mode perturbations.** Given a ring of cliques on 12 nodes (top left panel) with 12-dimensional node features (bottom left panel), we show our input data for the original and perturbed states of the graph structure (top row) and the features (bottom row).

turbations: *performance separability*, addressing **P1**, and *mode complementarity*, addressing **P2**.
**C3** **Experiments.** We demonstrate the value of our framework and measures through extensive experiments, focusing on real-world datasets with graph-level tasks.
**C4** **Recommendations.** Based on **C1**–**C3**, we derive concrete recommendations for improving benchmarking and evaluation practices in graph learning.

Our work opens new research directions in data-centric graph learning, and it constitutes a first step toward our long-term vision of enabling *evaluations of evaluations*: systematic assessments of the quality of evidence provided for the performance of new graph-learning methods.

**Structure.** In Section 2, we formally introduce RINGS, our mode-perturbation framework for evaluating graph-learning datasets, along with our proposed dataset quality measures. Having reviewed related work in Section 3, we demonstrate the practical utility of our framework through extensive experiments in Section 4. We discuss conclusions, limitations, and avenues for future work in Section 5. Detailed supplementaries are provided in Appendices A to E.

## 2. The RINGS Framework

After establishing our notation, we develop RINGS, our framework for evaluating graph-learning datasets. We do so in three steps, intuitively introducing and formally defining *mode perturbations* (2.1), *performance separability* (2.2), and *mode complementarity* (2.3).

**Preliminaries.** We work with attributed graphs $(G, X)$, where $G = (V, E)$ is a graph with $n = |V|$ nodes and $m = |E|$ edges, $X \subset \mathbb{R}^k$ is the space of $k$-dimensional node

features, and we assume w.l.o.g. that $|X| = n$. For graph-level tasks, we have datasets $\mathcal{D} = \{G_1, \ldots, G_N\}$, where $N$ is the total number of graphs. Given a set $S$, $2^S$ is its power set, and $\binom{S}{\ell}$ is the set of all $\ell$-element subsets of $S$. Multisets are denoted by $\{\{\cdot\}\}$, and the set of positive integers no greater than $\ell$ is written as $[\ell] = \{i \in \mathbb{Z} \mid 0 < i \leq \ell\}$.

## 2.1. Mode Perturbations

Given an attributed graph $(G, X)$, both data modes—i.e., the graph structure and the node features—are naturally associated with metric spaces that encode pairwise distances between nodes (i.e., distance matrices[2]). In our RINGS framework, we modify these metric spaces to reveal information about the quality of graph-learning datasets. This idea is formalized in the notion of *mode perturbation*.

**Definition 2.1** (Mode Perturbation)**.** A mode perturbation $\varphi$ is a map between attributed graphs such that $\varphi \colon (G, X) \mapsto (G', X')$.

This definition is very general, allowing $\varphi$, inter alia, to act on *both* the graph structure *and* the node features. For the purposes of understanding the connection between graph structure and node features in graph-learning datasets, however, we focus on mode perturbations that modify *either* the graph structure *or* the node features, as illustrated in Figure 1. We start by formalizing our *feature perturbations*.

**Definition 2.2** (Feature Perturbations)**.** Given an attributed graph $(G, X)$ on $n$ nodes with features $X \subset \mathbb{R}^k$, we define:

$$\varphi_{\mathrm{ef}} \colon (G, X) \mapsto (G, \mathbf{0}_n) \qquad \text{[empty features]}$$
$$\varphi_{\mathrm{cf}} \colon (G, X) \mapsto (G, \mathbf{I}_n) \qquad \text{[complete features]}$$
$$\varphi_{\mathrm{rf}} \colon (G, X) \mapsto (G, \mathcal{R}_{\mathrm{F}}(X)) \qquad \text{[random features]}$$

Here, $\mathcal{R}_{\mathrm{F}} \colon \mathbb{R}^{n \times k} \to \mathbb{R}^{n \times k'}$ randomizes the features $X$.

We can define a matching set of *structural perturbations* by modifying our original edge set.

**Definition 2.3** (Structural Perturbations)**.** Given an attributed graph $(G, X)$ with edge set $E$, we define:

$$\varphi_{\mathrm{eg}} \colon (G, X) \mapsto ((V, \emptyset), X) \qquad \text{[empty graph]}$$
$$\varphi_{\mathrm{cg}} \colon (G, X) \mapsto \left( \left( V, \binom{V}{2} \right), X \right) \quad \text{[complete graph]}$$
$$\varphi_{\mathrm{rg}} \colon (G, X) \mapsto ((V, \mathcal{R}_{\mathrm{S}}(E)), X) \qquad \text{[random graph]}$$

Here, $\mathcal{R}_{\mathrm{S}} \colon 2^{\binom{V}{2}} \to 2^{\binom{V}{2}}$ randomizes the edge set $E$.

For consistency, we also define the *original perturbation* $\varphi_{\mathrm{o}} \colon (G, X) \mapsto (G, X)$ [original].

---

[2]Deferring further details to Appendix A, we note that for finite metric spaces, we have $(Y, \mathrm{d}) \cong D \in \mathbb{R}^{n \times n}$, i.e., a finite matrix encoding pairwise distances between elements in $Y$. We use $D$ in contexts that rely on matrix operations, and $(Y, \mathrm{d})$ to emphasize both the original space and the associated metric.

To modify entire *collections* of graphs, we apply mode perturbations element-wise to all graphs in a given collection.

**Definition 2.4** (Dataset Perturbation)**.** Given a dataset $\mathcal{D}$ of attributed graphs and a mode perturbation $\varphi$, a *dataset perturbation* is given by $\varphi(\mathcal{D}) := \{\varphi(G, X) : (G, X) \in \mathcal{D}\}$.

## 2.2. Performance Separability

By systematically applying mode perturbations to a dataset $\mathcal{D}$, we create a *set of datasets* that contains several different versions of $\mathcal{D}$, capturing potentially interesting variation. This variation can be leveraged to investigate the properties derived in Section 1. First addressing **P1**, we now introduce *performance separability* as a measure to assess the extent to which both the graph structure and the node features of an attributed graph contain task-relevant information.

Intuitively, given two perturbations $\varphi, \varphi'$ of a dataset $\mathcal{D}$ as well as a task to be solved on $\mathcal{D}$, performance separability measures the *distance between performance distributions* associated with $\varphi(\mathcal{D})$ and $\varphi'(\mathcal{D})$. For a formal definition, we need notation describing these distributions.

**Definition 2.5** (Tuned Model)**.** A tuned model $\mathcal{M}$ is a triple $\mathcal{M} := (\mathcal{D}, \mathcal{A}, \theta)$, where $\mathcal{D}$ represents the dataset and associated task, $\mathcal{A}$ is the architecture used during training, and $\theta$ denotes the tuned parameters for the specific architecture.

To elucidate the relationship between the graph structure and the node features as it pertains to performance, within RINGS, we tune models not only on datasets $\mathcal{D}$ but also on perturbed datasets $\varphi(\mathcal{D})$.

**Definition 2.6** (Tuned Perturbed Model)**.** For a mode perturbation $\varphi$, $\mathcal{M}_{\varphi} := (\varphi(\mathcal{D}), \mathcal{A}, \theta)$ denotes a model tuned to solve the task associated with $\mathcal{D}$ under mode perturbation $\varphi$.

The performance distributions underlying our notion of performance separability can then be defined as follows.

**Definition 2.7** (Empirical Performance Distribution)**.** For a tuned (perturbed) model $\mathcal{M}_{\varphi}$, given an evaluation metric f and a set of initialization conditions $Z$, the *empirical performance distribution* $P_{\varphi}^{\mathrm{f}, Z}$ of $\mathcal{M}_{\varphi}$ is the distribution associated with performance measurements

$$\{\{\mathrm{f}(\mathcal{M}_{\varphi}; \zeta) \mid \zeta \in Z\}\} \, . \tag{1}$$

With *performance separability*, we now enable pairwise comparisons between the performance distributions of models trained and evaluated on distinct perturbations of $\mathcal{D}$.

**Definition 2.8** (Performance Separability)**.** Fix a dataset $\mathcal{D}$, an evaluation metric f, and initialization conditions $Z$, and let $\varphi, \varphi'$ be mode perturbations. We define the *performance separability* $\xi_{\mathrm{f}}^{\mathrm{Z}}$ of $\varphi$ and $\varphi'$ as

$$\xi_{\mathrm{f}}^{\mathrm{Z}}(\varphi, \varphi') := \mathbb{D}(P_{\varphi}^{\mathrm{f}, Z}, P_{\varphi'}^{\mathrm{f}, Z}) \, , \tag{2}$$

where $\mathbb{D}$ is a method comparing distributions.

In our experiments (Section 4.1), we use the Kolmogorov-Smirnov (KS) statistic with permutation testing to instantiate $\xi_f^Z$. This allows us to assess whether the performance distributions associated with $\mathcal{M}_\varphi$ and $\mathcal{M}_{\varphi'}$ are *significantly different*. To evaluate **P1**, we can then interpret a lack of (statistical) performance separability between a model trained on the original data and a model trained on a mode perturbation as evidence that the perturbed mode does not contain (non-redundant) task-relevant information.

### 2.3. Mode Complementarity

While performance separability allows us to assess dataset quality in a setting similar to traditional model evaluation, it has three main limitations. First, it is task-dependent due to its focus on **P1**, and thus risks underestimating datasets whose tasks are simply misaligned with the information contained in them. Second, it is measured based on, and hence still to some extent confounded by, model capabilities. And third, it is resource-intensive to compute. To address **P2** and forego these limitations, we propose *mode complementarity*.

Intuitively, as illustrated in Figure 2, for an attributed graph $(G, X)$, mode complementarity measures the distance between a metric space constructed from the graph structure and a metric space constructed from the node features. To formalize this, we define a process for constructing metric spaces from both modes in $(G, X)$ using lift functions.

**Definition 2.9** (Metric-Space Construction). For attributed graph $(G, X)$ and metric d, we construct metric spaces as

$$\mathcal{L}_d : (G, X) \mapsto (V, d),  \tag{3}$$

i.e., lifts that take in either *structure-based distances* arising from $G$ or *feature-based distances* arising from $X$ and produce a metric space over the node set $V$.

In RINGS, we combine these lifts with mode perturbations.

**Definition 2.10** (Perturbed Metric Space). Given an attributed graph $(G, X)$ and an associated metric d, the $\varphi$-perturbed metric space of $(G, X)$ under d is

$$D_d^\varphi := \mathcal{L}_d \circ \varphi (G, X),  \tag{4}$$

which we construe as a distance matrix.

This allows us to assess differences between mode perturbations by comparing metric spaces.

**Definition 2.11** (Metric-Space Comparison). For a fixed set of $n$ points and two metrics d and d$'$, we compare the metric spaces that arise from d and d$'$ by computing the $L_{p,q}$ norm of the difference of their $n \times n$ matrix representations, i.e.,

$$\mathcal{C}_{p,q}(D_d, D_{d'}) := \frac{\|D_d - D_{d'}\|_{p,q}}{\sqrt[q]{n^2 - n}}.  \tag{5}$$

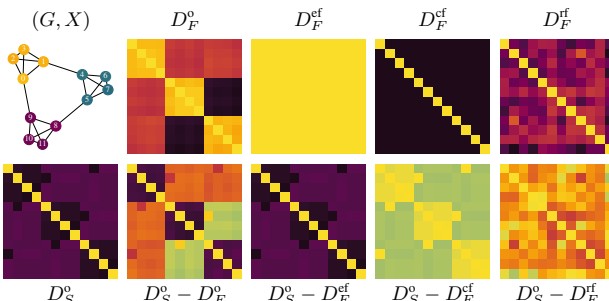

$(G, X)$    $D_F^o$    $D_F^{ef}$    $D_F^{cf}$    $D_F^{rf}$

$D_S^o$    $D_S^o - D_F^o$    $D_S^o - D_F^{ef}$    $D_S^o - D_F^{cf}$    $D_S^o - D_F^{rf}$

Figure 2: **Setup for mode-complementarity assessment.** For $(G, X)$ from Figure 1, we show the metric spaces arising from original graph structure ($D_S^o$), original node features ($D_F^o$), and 3 feature perturbations ($D_F^{ef}$, $D_F^{cf}$, $D_F^{rf}$), along with the differences between $D_S^o$ and the feature spaces that underlie our mode-complementarity computations. For structural perturbations, we swap the roles of $D_S^*$ and $D_F^*$. Note that the spaces arising from complete graph and complete (one-hot) features are equivalent, as are the (degenerate) spaces arising from empty graph and empty features.

With *mode complementarity*, we then measure the distance between the *normalized* metric spaces arising from the graph structure and the node features.

**Definition 2.12** (Mode Complementarity). Given an attributed graph $(G, X)$ with structural metric $d_S$, derived from $G$, and feature metric $d_F$, derived from $X$, we define the *mode complementarity* of $(G, X)$ as

$$\gamma^{p,q}(G, X) := \mathcal{C}_{p,q}(D_S, D_F),  \tag{6}$$

where $\mathcal{C}$ is a comparator from Definition 2.11,

$$D_S := D_{\overline{S}}/\text{diam}(D_{\overline{S}}) \text{ for } D_{\overline{S}} := \mathcal{L}_{d_S}(G, X),  \tag{7}$$
$$D_F := D_{\overline{F}}/\text{diam}(D_{\overline{F}}) \text{ for } D_{\overline{F}} := \mathcal{L}_{d_F}(G, X),  \tag{8}$$

using the lifts from Definition 2.9, and we leave (degenerate) zero-diameter spaces unchanged.

Note that since mode complementarity takes an $L_{p,q}$ norm of normalized metric spaces, we have $\gamma^{p,q}(G, X) \in [0, 1]$ by construction. To extend mode complementarity to mode perturbations, we again leverage function composition.

**Definition 2.13** (Perturbed Mode Complementarity). Given an attributed graph $(G, X)$ and a mode perturbation $\varphi$, the mode complementarity of $\varphi(G, X)$ is

$$\gamma^{p,q} \circ \varphi (G, X) = \mathcal{C}_{p,q}(D_S^\varphi, D_F^\varphi).  \tag{9}$$

For notational clarity, we use a subscript convention to denote the mode complementarity of $(G, X)$ under specific mode perturbations, i.e., $\gamma_*^{p,q}(G, X) := \gamma^{p,q} \circ \varphi_* (G, X))$.

To assess **P2**, we can interpret high *levels* of mode complementarity under a given mode perturbation as evidence that,

in the metric spaces associated with that perturbation, the graph structure and the features contain *complementary* information. We can also gain further insights by assessing the *differences* between mode-complementarity levels across mode perturbations. Depending on the mode perturbations compared, these differences reveal information about the nature of the connection between the graph structure and the node features, or about the diversity present in $G$ and $X$.

In Proposition A.10 (Appendix A), we formalize the relationship between empty mode perturbations $\varphi_{e*}$ and what we call *self-complementarity*, showing that $\gamma_{e*}^{p,q}(G, X)$ measures the geometric structure of the unperturbed mode. Based on both limiting behaviors of $\gamma_{e*}^{p,q}(G, X)$, which correspond to *uninformative* metric-space structures, we can then define a notion of *mode diversity*.

**Definition 2.14** (Mode Diversity). Given an attributed graph $(G, X)$, the *mode diversity* of $(G, X)$ for $* \in \{f, g\}$ is

$$\Delta_*^{p,q}(G, X) := 1 - |1 - 2\gamma_{e*}^{p,q}(G, X)| \in [0, 1] . \quad (10)$$

Intuitively, the mode diversity $\Delta_*^{p,q}$ scores the ability of $d_*$ to produce non-trivial geometric structure. Note that $\Delta_*^{p,q}(G, X) \to 0$ implies that $\mathcal{L}_{d_*} \to \mathbf{0}_n$ or $\mathcal{L}_{d_*} \to \mathbf{1}_n - \mathbf{I}_n$, our canonical uninformative metric spaces.

In our experiments, we instantiate $d_F$ with the Euclidean distance between node features and $d_S$ with a diffusion distance based on the graph structure (explained in Appendix B) to approximate the computations underlying graph-learning methods. We further choose the $L_{1,1}$ norm as our comparator, which yields a favorable duality, proved in Appendix A, between empty and complete perturbations.

**Theorem 2.15** (Perturbation Duality). *Fix an attributed graph $(G, X)$ and corresponding distances $d_S, d_F$ for lifting each mode into a metric space. For $* \in \{f, g\}$, Definition 2.12 of $\gamma$ yields the equivalence*

$$\gamma_{c*}^{1,1}(G, X) = 1 - \gamma_{e*}^{1,1}(G, X) . \quad (11)$$

## 3. Related Work

Here, we briefly contextualize our contributions, deferring a deeper discussion of related work to Appendix E.

Relating to *mode perturbations* and *performance separability*, Errica et al. (2020) create GNN experiments to improve reproducibility, and Bechler-Speicher et al. (2024) show that GNNs use the graph structure even when it is not conducive to a task (i.e., $\varphi_{eg}$ separably outperforms $\varphi_o$). RINGS is inspired by, and goes beyond, these works, providing a general framework for perturbation-based dataset evaluation.

Connecting with *mode complementarity*, researchers have been particularly interested in the effects of *homo- and heterophily* on node-classification performance (Lim et al.,

2021; Luan et al., 2023; Platonov et al., 2023a;b). While homophily characterizes the *task-dependent* relationship between graph structure and *node labels*, mode complementarity assesses the *task-independent* relationship between graph structure and *node features*. For node classification, Dong & Kluger (2023) propose the edge signal-to-noise ratio (ESNR), Qian et al. (2022) develop a subspace alignment measure (SAM), and Thang et al. (2022) analyze relations between *node features* and *graph structure* (FvS). However, with mode complementarity, we craft a score that (1) treats graph structure and node features as equal (unlike ESNR), (2) works on graphs without node labels and does not make assumptions about the spaces arising from edge connectivity and node features (unlike SAM, FvS), and (3) specifically informs graph-level learning tasks (unlike all of these works).

## 4. Experiments

We now demonstrate how to use RINGS to evaluate the quality of graph-learning datasets, guided by the principles developed in Section 1. Leveraging our mode-perturbation framework, we explore **P1** via performance separability and **P2** via mode complementarity, before distilling our observations into an actionable taxonomy of recommendations. For further details and additional results, including regression tasks, transformer architectures, and graph-level performance comparisons between models, see Appendix D.[3]

**Evaluated Datasets.** In our main experiments, we evaluate 13 popular graph-classification datasets: From the *life sciences*, we select AIDS, ogbg-molhiv (MolHIV), MUTAG, and NCI1 (small molecules), as well as DD, Enzymes, Peptides-func (Peptides), and PROTEINS-full (Proteins) (larger chemical structures). From the *social sciences*, we take COLLAB, IMDB-B, and IMDB-M (collaboration ego-networks), as well as REDDIT-B and REDDIT-M (online interactions). To approximate the standard evaluation setup as closely as possible and ensure compatibility with a wide range of models, for all datasets, we use the node features as encoded by PyTorch-geometric and disregard the edge features. See Appendix C for details and references.

### 4.1. Performance Separability (P1)

**Expectations.** (1) For a *structure- and feature-based task*, the original dataset should separably outperform all other mode perturbations. (2) For a *structure-based task*, the original dataset should separably outperform all structural perturbations. (3) For a *feature-based task*, the original dataset should separably outperform all feature perturbations. (4) If the original dataset is separably outperformed

---

[3]Our reproducibility package is publicly available via Zenodo at 10.5281/zenodo.15547322. The code is maintained on GitHub at https://github.com/aidos-lab/rings.

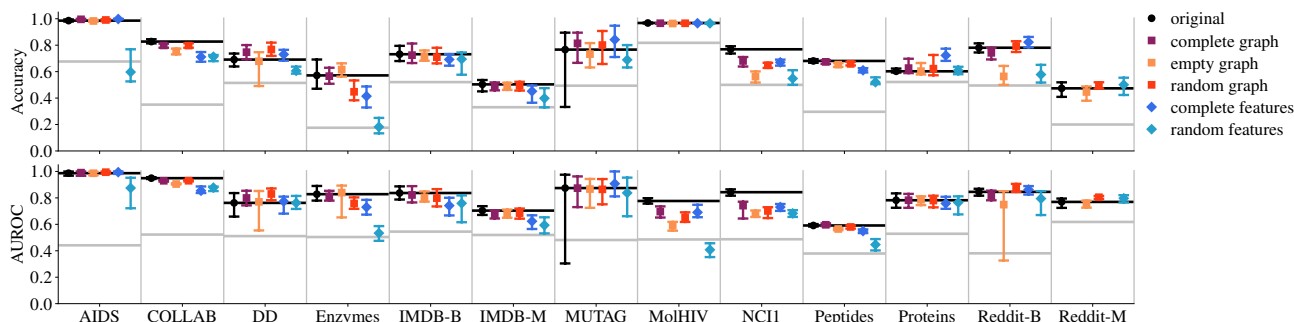

Figure 3: **Comparing *GNN performance* across different versions of the same dataset.** We show the mean (dot) and 95th percentile interval (bars) of accuracy and AUROC across 100 runs of the best (as measured by the respective performance mean) among our tuned GAT, GCN, and GIN models, for the original version and 5 perturbations of 13 graph-learning datasets. Black resp. silver horizontal lines show best mean trained resp. untrained performance on the original data. For Reddit-M, the complete-graph and complete-features perturbations failed to train due to memory problems, but the existing results already allow us to conclude that this dataset lacks performance separability. Note that AUROC is the recommended evaluation metric for MolHIV; we include accuracy here only for completeness.

| Dataset | Accuracy | AUROC | Structure | Features | Evaluation |
|---|---|---|---|---|---|
| AIDS | cf > cg > rg > o > eg > rf | cf/cg/rg > eg/o > rf | uninformative | uninformative | −− |
| COLLAB | o > cg/rg > eg > cf/rf | o > cg/rg > eg > rf > cf | informative | informative | ++ |
| DD | rg > cg > cf > eg/o > rf | rg > cf/cg/eg/o/rf | uninformative | uninformative | −− |
| Enzymes | eg > cg/o > rg > cf > rf | eg > o > cg > rg > cf > rf | uninformative | informative | − |
| IMDB-B | cf/cg/eg/o/rf/rg | o > cg > eg/rg > rf > cf | (un)informative | (un)informative | ∘ |
| IMDB-M | cg/eg/o/rg > cf > rf | o > cg/eg/rg > cf > rf | (un)informative | informative | + |
| MUTAG | cf/cg/o/rg > eg > rf | cf/o > cg/eg/rf/rg | (un)informative | uninformative | − |
| MolHIV | o > cf/cg/rg > rf > eg | o > cg > cf > rg > eg > rf | informative | informative | ++ |
| NCI1 | o > cg > cf > rg > eg > rf | o > cg > cf > rg > eg/rf | informative | informative | ++ |
| Peptides | o > cg > rg > eg > cf > rf | cg > o > rg > eg > cf > rf | (un)informative | informative | + |
| Proteins | cf > cg/rf > eg/o/rg | cf/cg/eg/o/rf/rg | uninformative | uninformative | −− |
| Reddit-B | cf > rg > o > cg > eg/rf | rg > cf > o > cg/eg/rf | uninformative | uninformative | −− |
| Reddit-M | rf > rg > o > eg | rg > rf > o > eg | uninformative | uninformative | −− |

Table 1: **Measuring *performance separability* between different versions of the same dataset.** To quantify the conclusions from Figure 3 and further account for performance *distributions* (Definition 2.7), we use permutation tests with 10 000 random permutations and the Kolmogorov-Smirnov (KS) statistic as our test statistic at an $\alpha$-level of 0.01, Bonferroni-corrected for multiple hypothesis testing within each individual dataset. Here, for sets $S_1$ and $S_2$, $S_1 > S_2$ denotes that the elements in the set $S_1$ separably outperform the elements in the set $S_2$ (i.e., the pairwise distances between $s_1$ and $s_2$ are statistically significantly different for all $s_1 \in S_1$ and $s_2 \in S_2$), and we represent the sets in condensed notation, concatenating elements with "/". We see that original datasets are often separably outperformed by their perturbed variants.

by a structural (feature) perturbation, the structural (feature) mode of the dataset is poorly aligned with the task.

**Measurement.** To evaluate performance separability, we train 3 standard GNN architectures (GAT, GCN, GIN) in the RINGS framework on the original and 5 perturbations of our 13 main datasets: {empty, complete, random} graph, and {complete, random} features.[4] With the setup described in Appendix Table 4, we tune and evaluate a total of 273 models. Although any GNN architecture could be used to evaluate performance separability within RINGS, using standard architectures allows us to focus on *dataset* evaluation.

For each version of each dataset, we identify the model with the best performance under evaluation metric f ∈ {accuracy, AUROC} as $(\mathcal{A}^\star, \theta^\star) := \arg\max_{(\mathcal{A},\theta)} \mathbb{E}\left[f(\mathcal{M}_\varphi; \zeta)\right]$, using the performance distribution of the top model, $\mathcal{M}_\varphi^\star := (\varphi(\mathcal{D}), \mathcal{A}^\star, \theta^\star)$, to assess performance separability.

To compare performance distributions, we use permutation tests with the Kolmogorov-Smirnov (KS) statistic, Bonferroni correcting for multiple hypothesis testing within each individual dataset and testing differences for significance at an $\alpha$-level of 0.01 (see Appendix Tables 10 and 11 for substantively identical results using different setups).

**Results.** We show the performance of the best GNN models

---

[4]GNNs cannot train with $\varphi_{\text{ef}}$ due to uninformative gradients.

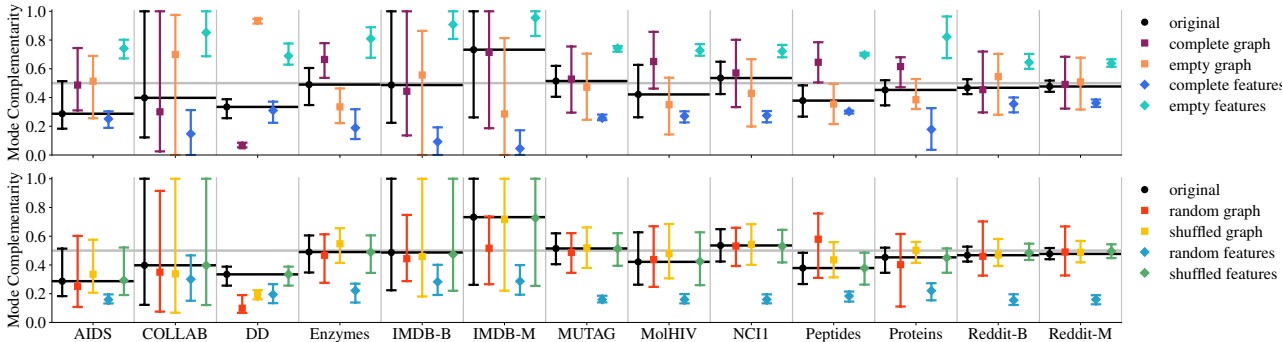

Figure 4: **Comparing *levels of mode complementarity* across different versions of the same dataset.** We show the mean (dot) and 95th percentile intervals (bars) of complementarity scores for the original version as well as 4 deterministic perturbations (top) and 4 randomized perturbations (bottom) of 13 graph-learning datasets, computed with $t = 1$ diffusion steps. Black horizontal lines indicate mean mode complementarities of the original dataset, and the silver horizontal line marks the 0.5 threshold relevant for assessing mode diversity. Note that $\gamma_{eg} = 1 - \gamma_{cg}$ and $\gamma_{ef} = 1 - \gamma_{cf}$ by Theorem 2.15.

in Figure 3, for the original and 5 mode perturbations over our 13 main datasets and 2 evaluation statistics, and summarize the associated statistical performance-separability results in Table 1. We find that only 3 datasets—COLLAB, MolHIV, and NCI1—satisfy the performance-separability relations expected from structure- and feature-based tasks (with Peptides coming close), whereas 5 datasets—AIDS, DD, Proteins, Reddit-B, and Reddit-M—do not satisfy *any* separability requirements, highlighting their low quality from the perspective of **P1**. Among our striking results, the original dataset is separably outperformed by (1) the random-graph perturbation on DD, (2) the empty-graph perturbation on Enzymes, and (3) the complete-features and random-graph perturbations on Reddit-B, indicating that the affected modes are poorly aligned with their current task.

### 4.2. Mode Complementarity (P2)

**Expectations.** (1) For a *structure- and feature-based task*, the original dataset should have high mode complementarity, and both modes should have high mode diversity (indicated by high mode complementarity of the empty and complete perturbations of their dual mode). (2) For a *structure-based task*, the structural mode should have high mode diversity. (3) For a *feature-based task*, the feature mode should have high mode diversity. (4) If the original dataset has low mode complementarity, the information contained in structure and features is redundant. (5) If both modes have low mode diversity, the dataset contains little insightful variation.

**Measurement.** We evaluate mode complementarity for the original as well as 4 fixed and 4 randomized mode perturbations: {complete, empty, random, shuffled} $\times$ {graph, features} (see Appendix D for detailed descriptions). To measure mode complementarity (and the mode diversity derived from it), for each graph in a given dataset pertur-

| Dataset | $\Delta_S$ | | $\Delta_F$ | | Structure | | Features | |
| --- | --- | --- | --- | --- | --- | --- | --- | --- |
| | $\mu$ | $\sigma$ | $\mu$ | $\sigma$ | $\mu$ | $\sigma$ | $\mu$ | $\sigma$ |
| AIDS | 0.52 | 0.07 | 0.81 | 0.14 | ∘ | − | ++ | ∘ |
| COLLAB | 0.30 | 0.20 | 0.27 | 0.24 | − | ++ | − | ++ |
| DD | 0.62 | 0.09 | 0.13 | 0.03 | ++ | − | −− | −− |
| Enzymes | 0.38 | 0.10 | 0.66 | 0.13 | − | − | ++ | ∘ |
| IMDB-B | 0.18 | 0.11 | 0.55 | 0.29 | −− | ∘ | ∘ | ++ |
| IMDB-M | 0.09 | 0.11 | 0.30 | 0.34 | −− | ∘ | − | ++ |
| MUTAG | 0.51 | 0.02 | 0.76 | 0.14 | ∘ | −− | ++ | ∘ |
| MolHIV | 0.55 | 0.05 | 0.69 | 0.21 | ∘ | −− | ++ | ++ |
| NCI1 | 0.56 | 0.05 | 0.78 | 0.16 | ∘ | −− | ++ | ++ |
| Peptides | 0.61 | 0.01 | 0.71 | 0.23 | ++ | −− | ++ | ++ |
| Proteins | 0.36 | 0.12 | 0.76 | 0.08 | − | ∘ | ++ | − |
| Reddit-B | 0.71 | 0.05 | 0.80 | 0.12 | ++ | − | ++ | ∘ |
| Reddit-M | 0.72 | 0.03 | 0.85 | 0.11 | ++ | −− | ++ | ∘ |

Table 2: **Evaluating mode diversity.** We assess the mean ($\mu$) and standard deviation ($\sigma$) of structural diversity ($\Delta_S$) and feature diversity ($\Delta_F$) for 13 graph-learning datasets.

bation, we instantiate Definition 2.12 using the Euclidean distance as our feature distance, the diffusion distance (see Appendix B) for a number of diffusion steps $t \in [10]$ as our graph distance, and the $L_{1,1}$ norm as our comparator (see Appendix B for examples of other choices).

**Results.** In Figure 4, we show the mean and 95th percentile intervals of the mode-complementarity distributions at $t = 1$, for the original and the 8 other selected mode perturbations, for each of our 13 main datasets. We observe interesting differences between mode-complementarity profiles across datasets. For example, (1) COLLAB, IMDB-B, and IMDB-M stand out for their large ranges (likely due to their nature as ego-networks); (2) DD attains extreme values with its complete-graph and empty-graph perturbations (likely due to the absence of features in the original dataset, translated into complete-type features by PyTorch-

geometric); and (3) Peptides has mode complementarities in the random-graph and the complete-graph perturbations that are noticeably higher than that of the original dataset (possibly related to the presence of long-range connections).

Supplementing Figure 4 with mode-diversity scores in Table 2, we find that feature diversity is more common than structural diversity, and variation in structural diversity is almost always low. Notably, (1) COLLAB, IMDB-B, and IMDB-M, judged favorably or neutral by performance separability, score poorly on structural and feature diversity, suggesting that their tasks may not reveal much information about the power of graph-learning methods; (2) conversely, among the datasets judged problematic by performance separability, DD exhibits high levels of structural diversity, indicating its potential for adequately aligned structural tasks; and (3) Reddit-B and Reddit-M show high levels of structural and feature diversity, indicating untapped potential for graph-learning tasks that are based on both structure *and* features, as well as high quality from the perspective of **P2**.

To conclude, we explore the relationship between mean mode complementarity and mean (AUROC) performance across all mode perturbations in Figure 5 (see Appendix Figure 10 and Table 18 for extended results). We find that higher mode complementarity is generally associated with higher performance, with exceptions for some tasks we previously identified as misaligned (DD and Reddit-M). Since evaluating performance separability is comparatively costly, this further underscores the value of mode complementarity as a task-independent diagnostic for graph-learning datasets.

### 4.3. Dataset Taxonomy

From our observations on performance separability (**P1**) and mode complementarity (**P2**), we can distill the following actionable dataset taxonomy, where we note evidence from performance separability (†) and mode complementarity (‡):

| Action | Datasets |
|---|---|
| Keep (†\|‡) | MolHIV, NCI1, Peptides |
| Realign (‡) | AIDS, DD, MUTAG, Reddit-B, Reddit-M |
| Deprecate (‡) | COLLAB, IMDB-B, IMDB-M |
| Deprecate (†\|‡) | Enzymes, Proteins |

**Taxonomy, categories, and interpretation.** Our taxonomy is built on our measures of performance separability and structural diversity (derived from mode complementarity). Here, we emphasize structural diversity over feature diversity because the structural mode is characteristic of the graph-learning setting, acknowledging that datasets with high feature diversity could still be useful as benchmarks in settings outside of graph learning. Here, we call the value of a measure *high* if it is judged as ∘, +, or ++ (see Tables 1 and 2), and *low* otherwise. In the following, we elaborate

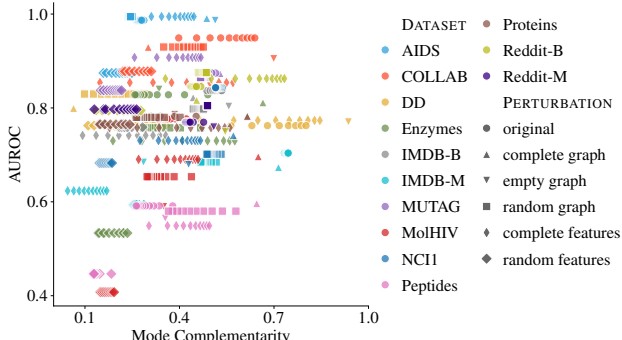

Figure 5: **Mode complementarity and performance.** We show mean AUROC (y) as a function of mean mode complementarity (x), for the original version and 5 perturbations of our 13 main datasets, based on our best-on-average models (as in Figure 3). Each marker represents a (dataset, perturbation, $t$) tuple, where $t \in [10]$ is the number of diffusion steps in the diffusion distance. Higher mean mode complementarity is associated with higher AUROC, and datasets differ in the range of their mode-complementarity shifts.

on the definition and interpretation of each action category.

*Keep (†\|‡).* This category contains datasets with *high* performance separability and *high* structural diversity. These datasets constitute valuable graph-learning benchmarks.

*Realign (‡).* We call a dataset *misaligned* if it exhibits *low* performance separability but *high* structural diversity. *Realignment*, then, collectively denotes several potential operations, including changing the prediction targets (e.g., using different categories to classify discussion threads in Reddit-B or Reddit-M) or changing the prediction task (e.g., moving from a graph-level to a node-level task).

*Deprecate (‡).* This category contains datasets with *high* performance separability but *low* structural diversity. Although both structure and features may be needed to achieve state-of-the-art performance, these datasets do not contain interesting structural variation, and as such, they do little to probe the capabilities of graph-learning models.

*Deprecate (†\|‡).* This category contains datasets that exhibit *low* structural diversity and *low* performance separability. These datasets do not currently require the capabilities of graph-learning models to integrate information from structure and features, and they also do not hold sufficient structural diversity to change this via realignment.

**Separability, complementarity, and misalignment.** While our experiments show that higher mode complementarity is associated with higher performance (see Figure 4), we make no claims about the relationship between mode diversity and performance separability. Since performance separability assesses the task-specific performance gap between a dataset

and its perturbations, and mode diversity measures the task-agnostic variation contained in the modes of an individual dataset, we would not expect high mode diversity to be consistently associated with high performance separability. For example, a dataset could have high structural and feature diversity but low mode complementarity, which could lead to *complete graph* and *complete features* performing on par with the *original* mode, eliminating separability.

Misalignment occurs when there is interesting variation in the data that models could leverage in their predictions (high mode diversity), but the existing relationship between this variation and the prediction target does not provide a significant performance advantage over settings in which this relationship is deliberately destroyed (lack of performance separability). While the lack of performance separability could also be due to limitations of the models employed (i.e., they may not be expressive enough), given our comprehensive measurement setup, we deem it more likely that the relationship between the variation in the data and the prediction target is strained, prompting the need for realignment. However, more work is needed to disentangle the model-related and the data-related factors contributing to a lack of performance separability.

## 5. Discussion

**Conclusions.** We introduced RINGS, a principled framework for evaluating graph-learning datasets based on *mode perturbations*—i.e., controlled changes to their graph structure and node features. In RINGS, we developed *performance separability* as a task-dependent, and *mode complementarity* as a task-independent measure of dataset quality. Using our framework to categorize 13 popular graph-classification datasets, we identified several benchmarks that require realignment (e.g., by better data modeling) or complete deprecation.

While some datasets have been observed to be problematic in prior work, with RINGS, we offer a full perturbation-based evaluation pipeline. We designed our initial set of perturbations from a data-centric perspective, but additional, cleverly crafted perturbations could also isolate and confirm the performance separability of new model contributions. Thus, we hope that RINGS will raise the standard of proof for model-centric evaluation in graph learning as well.

**Limitations.** Given that message passing is the predominant paradigm in graph learning, our *performance-separability* experiments mostly targeted message-passing GNNs, at the exclusion of other architectures. While RINGS is a general framework, our current norm-based comparison of metric spaces assumes that nodes and node features are *paired*. Thus, we presently do not measure aspects like the *metric distortion* arising from different mappings. Moreover, our

notion of *mode complementarity* is task-independent and model-agnostic. This allows us to gain insights into *datasets*, but the inclusion of additional information about models would enable claims about (*model*, *dataset*, *task*) triples.

**Future Work.** We envision RINGS to support the design of better datasets and more challenging tasks for graph learning, and we hope that the community will use our measures to further scrutinize its evaluation infrastructure. Additionally, we highlight four concrete avenues for future work.

**F1 Theoretical analysis.** While we provided initial theoretical results on the behavior of perturbed mode complementarity, a comprehensive theoretical analysis of its properties is yet outstanding. Adopting an information-theoretic perspective could yield valuable insights.

**F2 Task coverage.** Our work focused on graph-level tasks, but node-level tasks and edge-level tasks, too, merit closer inspection. While performance separability immediately applies to such tasks, extending mode complementarity requires further conceptual advances. These advances might also naturally give rise to a *localized* notion of mode complementarity.

**F3 Feature coverage and graph coverage.** RINGS naturally accounts for node features, but some datasets also contain edge features or graph-level features that could be incorporated into our framework. Likewise, defining mode complementarity for more expressive types of graphs—from temporal graphs to hypergraphs—would constitute a valuable extension of RINGS.

**F4 Model interpretability.** While RINGS was designed for dataset evaluation, it can also enhance the interpretability of graph-learning models. For example, one could study how mode complementarity changes during training, or how mode-complementarity distributions impact train-eval-test splits. This could yield interesting insights into the relationship between mode complementarity and performance, and it could help elucidate the inner workings of individual models.

Finally, RINGS could support the design of better graph-learning models. As highlighted by our dataset taxonomy, by identifying datasets that do not require the specific capabilities of graph-learning models, performance separability helps direct model-development resources to where they are most needed. Moreover, mode complementarity and mode diversity could guide architectural choices for a specific task on a specific dataset (e.g., whether a model should focus on leveraging the graph structure or the node features). These measures could also inform model designs that incorporate data-centric components to enhance performance adaptively (both at the level of individual observations and at the level of entire datasets)—e.g., by preprocessing the data to increase mode complementarity. However, more research is needed to realize the potential of RINGS in these domains.

## Acknowledgments

This work has received funding from the Swiss State Secretariat for Education, Research, and Innovation (SERI). We acknowledge the use of the High Performance Computing cluster at Helmholtz Munich, maintained by the Scientific Computing and DigIT teams, as well as the computational resources provided by the Aalto Science-IT project.

## Impact Statement

With RINGS, we introduce a principled data-centric framework for evaluating the quality of graph-learning datasets. We expect our framework to help improve data and evaluation practices in graph learning, enabling the community to focus on datasets that truly require resource-intensive graph-learning approaches via mode complementarity and encouraging greater experimental rigor via performance separability. Thus, we hope to contribute to reducing the environmental impact of the machine-learning community, and to the development of scientifically sound, responsible AI methods.

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

# Appendix

In this appendix, we provide the following supplementary materials.

## A. Extended Theory

### A.1. Metric Spaces

Here, we introduce metric spaces as well as various related notions that improve our understanding of *mode complementarity*.

**Definition A.1** (Metric Space). A *metric space* is a tuple $(Y, \mathrm{d})$ that consists of a nonempty set $Y$ together with a function $\mathrm{d} \colon Y \times Y \to \mathbb{R}$ which satisfies the axioms of a metric, i.e., (i) $\mathrm{d}(x, y) \geq 0$ for all $x, y \in Y$ and $\mathrm{d}(x, y) = 0$ if and only if $x = y$, (ii) $\mathrm{d}(x, y) = \mathrm{d}(y, x)$, and (iii) $\mathrm{d}(x, y) \leq \mathrm{d}(x, z) + \mathrm{d}(z, y)$.

**Definition A.2** (Isometry). Given two metric spaces $(X, \mathrm{d}_X)$ and $(Y, \mathrm{d}_Y)$, we call a function $f \colon X \to Y$ an *isometry* if $\mathrm{d}_Y(f(x), f(y)) = \mathrm{d}_X(x, y)$ for all $x, y \in X$.

**Definition A.3** (Trivial Metric Space). We define a *trivial metric space* as any metric space $(Y, \mathrm{d})$ in the equivalence class of $\{\mathrm{x}\}$ (the single point space) under isometry. In particular, this corresponds to any finite space paired with a metric d such that $\mathrm{d}(x, y) = 0$ for all $x, y$. For finite spaces of size $n$, we represent this as the $n \times n$ matrix of zeros, $\mathbf{0}_n$.

**Definition A.4** (Discrete Metric Space). A metric space $(Y, \mathrm{d})$ is a *discrete metric space* if d satisfies

$$d(x, y) = \begin{cases} 1, & \text{if } x \neq y \\ 0, & \text{otherwise .} \end{cases} \tag{12}$$

For finite spaces of size $n$, we represent this as the $n \times n$ matrix of all ones with zero diagonal, $\mathbf{1}_n - \mathbf{I}_n$. Note that when normalizing non-degenerate spaces to unit diameter, the discrete metric space is one where all non-equal elements are maximally distant from each other, (e.g. a complete graph).

We go from attributed graphs to metric spaces using the lift construction presented in the main paper, restated here for completeness.

**Definition 2.9** (Metric-Space Construction). For attributed graph $(G, X)$ and metric d, we construct metric spaces as

$$\mathcal{L}_{\mathrm{d}} \colon (G, X) \mapsto (V, \mathrm{d}), \tag{3}$$

i.e., lifts that take in either *structure-based distances* arising from $G$ or *feature-based distances* arising from $X$ and produce a metric space over the node set $V$.

### A.2. Perturbations

For convenience in referencing our theory, we repeat our definitions here:

**Definition 2.1** (Mode Perturbation). A mode perturbation $\varphi$ is a map between attributed graphs such that $\varphi \colon (G, X) \mapsto (G', X')$.

**Definition 2.3** (Structural Perturbations). Given an attributed graph $(G, X)$ with edge set $E$, we define:

$$\varphi_{\mathrm{eg}} \colon (G, X) \mapsto ((V, \emptyset), X) \qquad \text{[empty graph]}$$
$$\varphi_{\mathrm{cg}} \colon (G, X) \mapsto \left( \left( V, \binom{V}{2} \right), X \right) \quad \text{[complete graph]}$$
$$\varphi_{\mathrm{rg}} \colon (G, X) \mapsto ((V, \mathcal{R}_{\mathrm{S}}(E)), X) \qquad \text{[random graph]}$$

Here, $\mathcal{R}_{\mathrm{S}} \colon 2^{\binom{V}{2}} \to 2^{\binom{V}{2}}$ randomizes the edge set $E$.

**Definition 2.2** (Feature Perturbations). Given an attributed graph $(G, X)$ on $n$ nodes with features $X \subset \mathbb{R}^k$, we define:

$$\varphi_{\mathrm{ef}} \colon (G, X) \mapsto (G, \mathbf{0}_n) \qquad \text{[empty features]}$$
$$\varphi_{\mathrm{cf}} \colon (G, X) \mapsto (G, \mathbf{I}_n) \qquad \text{[complete features]}$$
$$\varphi_{\mathrm{rf}} \colon (G, X) \mapsto (G, \mathcal{R}_{\mathrm{F}}(X)) \qquad \text{[random features]}$$

Here, $\mathcal{R}_{\mathrm{F}} \colon \mathbb{R}^{n \times k} \to \mathbb{R}^{n \times k'}$ randomizes the features $X$.

**Definition A.5** (Randomization Methods). Assume we have $n$ nodes in $(G, X)$. Let $\pi \colon S \to S$ be a permutation over an ordered set.

$$\mathcal{R}_{\mathrm{F},1}(X, k) \mapsto \{\mathbf{x_i} \sim \mathcal{N}(\mathbf{0}, \mathbf{I}_k)\} \quad \text{[random features]}$$
$$\mathcal{R}_{\mathrm{F},2}(X) \coloneqq \pi(X) \qquad \text{[shuffled features]}$$
$$\mathcal{R}_{\mathrm{S},1}(G, p) \coloneqq ER(n, p) \qquad \text{[random graph]}$$
$$\mathcal{R}_{\mathrm{S},2}(G) \coloneqq (V, \pi(E)) \qquad \text{[shuffled graph]}$$

where $ER(n, p)$ represents an *Erdős–Rényi* model, ensuring that each edge $(u, v) \in \binom{V}{2}$ is included in $E$ independently with probability $p$.

### A.3. Complementarity

**Definition 2.10** (Perturbed Metric Space). Given an attributed graph $(G, X)$ and an associated metric d, the $\varphi$-perturbed metric space of $(G, X)$ under d is

$$D_{\mathrm{d}}^{\varphi} \coloneqq \mathcal{L}_{\mathrm{d}} \circ \varphi (G, X), \tag{4}$$

which we construe as a distance matrix.

### A.3.1. COMPARATORS

**Definition A.6** ($L_{p,q}$ Norm). For an $n \times n$ pairwise distance matrix $D$ representing a finite metric space $(Y, \mathrm{d})$, we define the $L_{p,q}$ norm as

$$||D||_{p,q} = \left( \frac{\sum_{i=1}^{n} \left( \sum_{j=1}^{n} |\mathrm{d}(i,j)|^p \right)^{q/p}}{n(n-1)} \right)^{1/q} . \quad (13)$$

**Definition 2.11** (Metric-Space Comparison). For a fixed set of $n$ points and two metrics $\mathrm{d}$ and $\mathrm{d}'$, we compare the metric spaces that arise from $\mathrm{d}$ and $\mathrm{d}'$ by computing the $L_{p,q}$ norm of the difference of their $n \times n$ matrix representations, i.e.,

$$\mathcal{C}_{p,q}(D_{\mathrm{d}}, D_{\mathrm{d}'}) := \frac{||D_{\mathrm{d}} - D_{\mathrm{d}'}||_{p,q}}{\sqrt[q]{n^2 - n}} . \quad (5)$$

**Definition 2.12** (Mode Complementarity). Given an attributed graph $(G, X)$ with structural metric $\mathrm{d}_S$, derived from $G$, and feature metric $\mathrm{d}_F$, derived from $X$, we define the *mode complementarity* of $(G, X)$ as

$$\gamma^{p,q}(G, X) := \mathcal{C}_{p,q}(D_S, D_F) , \quad (6)$$

where $\mathcal{C}$ is a comparator from Definition 2.11,

$$D_S := D_{\overline{S}}/\mathrm{diam}(D_{\overline{S}}) \ \text{ for } \ D_{\overline{S}} := \mathcal{L}_{\mathrm{d}_S}(G, X) , \quad (7)$$
$$D_F := D_{\overline{F}}/\mathrm{diam}(D_{\overline{F}}) \ \text{ for } \ D_{\overline{F}} := \mathcal{L}_{\mathrm{d}_F}(G, X), \quad (8)$$

using the lifts from Definition 2.9, and we leave (degenerate) zero-diameter spaces unchanged.

### A.3.2. COMPLEMENTARITY UNDER PERTURBATION

**Definition 2.13** (Perturbed Mode Complementarity). Given an attributed graph $(G, X)$ and a mode perturbation $\varphi$, the mode complementarity of $\varphi(G, X)$ is

$$\gamma^{p,q} \circ \varphi (G, X) = \mathcal{C}_{p,q}(D_S^{\varphi}, D_F^{\varphi}) . \quad (9)$$

**Definition A.7** (Complementarity for Disconnected Graphs). For a disconnected attributed graph $(G, X)$ with $n$ nodes and $C$ connected components, we write $G$ as a union $\bigcup G_i$. Let $X|_{V_i}$ denote the subset of features corresponding to nodes in $G_i = (V_i, E_i)$. We define the *complementarity for disconnected graphs* as the weighted average of the individual component scores

$$\gamma_{\varphi}^{p,q}(G, X) := \sum_{i}^{C} \frac{n_i}{n} \left( \gamma_{\varphi}^{p,q}(G_i, X|_{V_i}) \right) , \quad (14)$$

where $n_i = |V_i|$. Axiom (ii) in Definition A.1, implies that isolated nodes are then assigned a trivial metric space.

**Lemma A.8** (Empty Perturbations Lift to Trivial Metric Spaces). *Let $(G, X)$ be an attributed graph. For any metric $\mathrm{d}_*$, the image of $\mathcal{L}_{\mathrm{d}_*}$ as defined in Definition 2.9 under precomposition with $\varphi_{e*}$ is a trivial metric space, for $* \in \{f, g\}$.*

*Proof.* $(* = f)$ By Definition 2.2, we have $\varphi_{\mathrm{ef}}(G, X) = (G, \mathbf{0}_n)$. Since $\mathbf{0}_n$ is isometric to $\{\mathrm{x}\}$, any metric must lift $\{\mathrm{x}\}$ to a trivial metric space (by metric-space axiom (ii)).

$(* = g)$ For structural metrics $\mathrm{d}_S$, we restrict to those derived from the adjacency matrix $A$ of $G$. By convention, $(V, \emptyset)$ is represented as $\mathbf{0}_n$. Hence, an identical argument holds *mutatis mutandis*, as any metric derived from the zero matrix lifts to a trivial metric space. $\square$

**Lemma A.9** (Complete Perturbations Lift to Discrete Metric Spaces). *Let $(G, X)$ be an attributed graph. For any choice of $\mathrm{d}_*$, the image of $\mathcal{L}_{\mathrm{d}_*}$ as defined in Definition 2.9 under precomposition with $\varphi_{c*}$ is a discrete metric space, for $* \in \{f, g\}$.*

*Proof.* $(* = f)$ By Definition 2.2, we have $\varphi_{\mathrm{ef}}(G, X) = (G, \mathbf{I}_n)$. The identity matrix $\mathbf{I}_n$ consists of standard basis vectors $e_i$ (i.e., the one-hot encodings of nodes). We claim that any metric $\mathrm{d}_F$ over $\mathbf{I}_n$ must be discrete.

Toward a contradiction, assume otherwise. W.l.o.g., suppose the metric space is scaled to unit diameter, implying that there exist $i, j$ such that $\mathrm{d}_F(e_i, e_j) < 1$. Since $(\mathbf{I}_n, \mathrm{d}_F)$ is uniform, it is invariant under orthogonal transformations, meaning we can permute the standard basis vectors while preserving distances. Thus, for all $k \neq i$, we have $\mathrm{d}_F(e_i, e_k) = \mathrm{d}_F(e_i, e_j) < 1$, contradicting our assumption. Therefore, $\mathrm{d}_F(x, y) = 1$ for all $x \neq y$, proving discreteness.

$(* = g)$ Here, $\varphi_{\mathrm{cg}}(G, X) = \left( V, \binom{V}{2} \right)$. The edge set consists of all two-element subsets of $V$, forming a fully connected graph. Similar to Lemma A.8, we restrict to metrics defined over the adjacency matrix $A$ of $G$. By construction, $A = \mathbf{1}_n - \mathbf{I}_n$, which has the same structure as $\mathbf{I}_n$ under the isometry $f : x_i \mapsto e_i$. Since all elements are maximally distant, any metric $\mathrm{d}_S$ must lift to a discrete space. $\square$

**Theorem 2.15** (Perturbation Duality). *Fix an attributed graph $(G, X)$ and corresponding distances $\mathrm{d}_S, \mathrm{d}_F$ for lifting each mode into a metric space. For $* \in \{f, g\}$, Definition 2.12 of $\gamma$ yields the equivalence*

$$\gamma_{c*}^{1,1}(G, X) = 1 - \gamma_{e*}^{1,1}(G, X) . \quad (11)$$

*Proof.*

$$\gamma_{cg}^{1,1}(G, X) \stackrel{\text{Def. 2.13}}{=} \mathcal{C}_{1,1}(D_{\mathrm{d}_S}^{\varphi_{\mathrm{cg}}}, D_{\mathrm{d}_F})$$

$$\stackrel{\text{Lem. A.9}}{=} \mathcal{C}_{1,1}(\mathbf{1}_n - \mathbf{I}_n, D_{\mathrm{d}_F})$$

$$\stackrel{\text{Def. 2.11}}{=} ||(\mathbf{1}_n - \mathbf{I}_n) - D_{\mathrm{d}_F}||_{1,1}$$

$$\stackrel{\text{Def. A.6}}{=} \frac{1}{n^2 - n} \left( \sum_{i=1}^{n} \sum_{j \neq i}^{n} |1 - \mathrm{d}_F(i, j)| \right)$$

$$\overset{d_F \leq 1}{=} \frac{1}{n^2 - n} \left( n^2 - n - \sum_{i=1}^{n} \sum_{j \neq i}^{n} d_F(i,j) \right)$$

$$\overset{\text{Def. A.1(ii)}}{=} 1 - \frac{1}{n^2 - n} \left( \sum_{i=1}^{n} \sum_{j=1}^{n} d_F(i,j) \right)$$

$$\overset{\text{Def. A.6}}{=} 1 - \|D_{d_F}\|_{1,1}$$

$$\overset{\text{Def. 2.13, Lem. A.8}}{=} 1 - \gamma_{eg}^{1,1}(G,X) \qquad \qquad \square$$

A symmetrical argument can be applied to $\gamma_{cf}^{1,1}$, concluding the proof.

**Definition 2.14** (Mode Diversity). Given an attributed graph $(G,X)$, the *mode diversity* of $(G,X)$ for $* \in \{f, g\}$ is

$$\Delta_*^{p,q}(G,X) := 1 - |1 - 2\gamma_{e*}^{p,q}(G,X)| \in [0,1] . \quad (10)$$

**Proposition A.10** (Self-Complementarity). *Let $d_S$ and $d_F$ be metrics over the modes of an attributed graph $(G,X)$. We define $(\varphi_{eg}, d_F)$ and $(\varphi_{ef}, d_S)$ as dual pairs that can be used to analyze the self-complementarity of a single mode. By construction, for $* \in \{f, g\}$, we have:*

$$\mathcal{L} \circ \varphi_{e*}(G,X) \triangleq \mathbf{0}_n . \quad (15)$$

*Thus, the complementarity under empty perturbations, $\gamma_{e*}^{p,q}$, measures the internal structure of the nonzero dual mode:*

$$\gamma_{e*}^{p,q}(G,X) = \frac{1}{\sqrt[q]{n^2 - n}} \|D_{d_*}\|_{p,q}. \quad (16)$$

*Due to the normalization, we obtain $\gamma_{e*}^{p,q}(G,X) \in [0,1]$. The limiting behavior of $\gamma_{e*}^{p,q}(G,X)$, then, provides insights into the underlying metric structure of the dual mode:*

*(1) If $\gamma_{e*}^{p,q}(G,X) \to 0$, then $d_*(x,y) \approx 0$ for all $x, y$.*
*(2) If $\gamma_{e*}^{p,q}(G,X) \to 1$, then $d_*(x,y) \approx 1$ for all $x \neq y$.*

# B. Extended Methods

## B.1. Mode Complementarity

The perturbations introduced with our RINGS framework are specifically designed to disentangle the relationship between the structural mode and the feature mode of $(G,X)$. Unlike existing methods, RINGS allows us to leverage the relationships both *between* and *within* individual modes:

$$
\begin{array}{ccc}
G & \longleftrightarrow & G' \\
\uparrow & & \uparrow \\
\downarrow & & \downarrow \\
X & \longleftrightarrow & X'
\end{array}
$$

Here, all arrows indicate the calculation of potential similarity measures. Solid arrows describe settings for which similarity measures are already known, such as *graph kernels* (Borgwardt et al., 2020; Kriege et al., 2020), *graph edit distances* (Gao et al., 2010), or distance metrics on $\mathbb{R}^n$. The dashed arrows show our ability to study *complementarity* and *self-complementarity* using metric spaces and principled mode perturbations.

## B.2. Metric Choices

As outlined in Section 2.3, we would like to understand, for a given attributed graph $(G,X)$ the complementarity between the information in the graph structure and the node features. Thus, we use $\gamma(G,X)$ to score the difference in geometric information contained in the two modes. As per Definition 2.12, this requires a choice of a *structural metric* ($d_S$) and a *feature metric* ($d_F$), which facilitate lifting the modes into a metric space.

While our experimental results are based on diffusion distance as our structural distance metric ($d_S$) and Euclidean distance in $\mathbb{R}^k$ as our feature-based distance metric ($d_F$), it is worth noting that the RINGS framework generalizes to other distance metrics. Below, we define the diffusion distance and Euclidean distance, as well as several other distance metrics suitable for the RINGS framework. To build intuition, we additionally depict the behavior of different graph-distance candidates under edge addition on toy data in Figure 6 and illustrate the interplay between metric choices for the real-world Peptides dataset in Figure 7.

### B.2.1. STRUCTURAL METRICS

**Disconnected Graphs** Here we define the metrics we use to lift a graph structure into a metric space. Although common practice assigns an infinite distance between unconnected nodes, due to our treatment of disconnected graphs in Definition A.7, we only lift metric spaces within a connected component. Our treatment of isolated nodes assigns trivial metric spaces to each node, and their union we also treat as trivial as seen in Lemma A.8.

**Definition B.1** (Laplacian Variants). We define the Laplacian of $G = (V, E)$ as

$$L := D - A , \quad (17)$$

where $A$ is the $n \times n$ adjacency matrix specified by $E$, and $D$ is the degree matrix. We also define a *normalized* Laplacian

$$\widehat{L} := D^{-1/2} L D^{-1/2} , \quad (18)$$

where $D^{-1/2} = \text{diag}\left( \frac{1}{\sqrt{D_{ii}}} \right)$.

**Definition B.2** (Diffusion Distance [$d_{S_t}$] (Coifman & Lafon, 2006)). We define our distance over $G$ for $t$ *diffusion steps* as

$$d_{S_t}(x,y) = \|\Psi_t(x) - \Psi_t(y)\| , \quad (19)$$

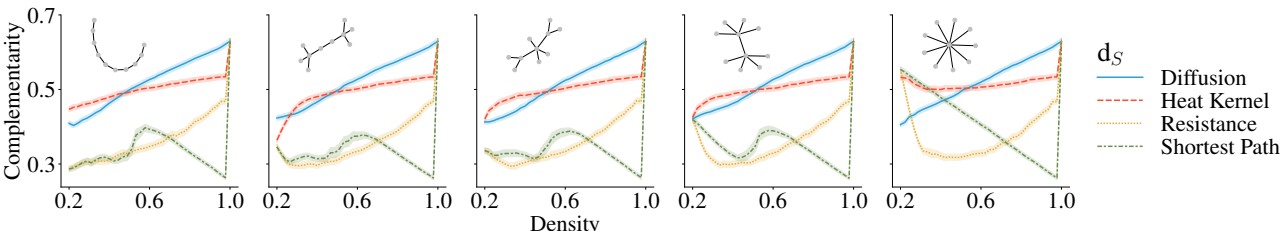

Figure 6: **Mode complementarity under different graph distances and varying density.** We show the evolution of mode complementarity for 5 graphs on 10 nodes with normally distributed one-dimensional features as we randomly add edges to increase the density from the minimum for a connected graph (0.2) to the maximum (1.0), for different choices of graph distance $d_S$. For the diffusion distance and the heat-kernel distance, which depend on the parameter $t$, we show the complementarity at $t = 1$. The diffusion distance exhibits the smoothest behavior.

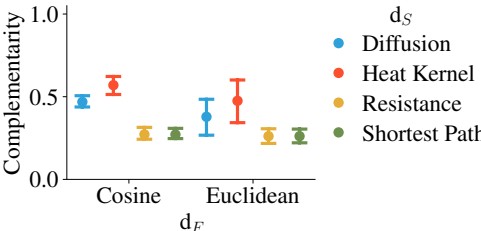

Figure 7: **Real-world mode complementarity under different graph distances $d_S$ and feature distances $d_F$.** We show the 95th percentile of the mode-complementarity distribution for the Peptides dataset, for varying combinations of graph distance $d_S$ and feature distance $d_F$, using $t = 1$ for the $t$-dependent distances (diffusion and heat kernel).

where $\{\Psi_t\}$ is a family of diffusion maps computed from the spectrum of $\widehat{L}$. In particular,

$$\Psi_t(x) \triangleq \begin{pmatrix} \lambda_1^t \psi_1(x) \\ \lambda_2^t \psi_2(x) \\ \vdots \\ \lambda_s^t \psi_s(x) \end{pmatrix},$$

where $\lambda_i, \psi_i$ are the $i^{th}$ eigenvalue and eigenvector of $\widehat{L}$.

**Definition B.3** (Heat-Kernel Distance (Bai & Hancock, 2004)). To compute the heat-kernel distance between nodes in a graph $G$, we take the spectral decomposition of $\widehat{L}$ as

$$\widehat{L} = \Psi \Lambda \Psi^T = \sum_{i=1}^{n} \lambda_i \psi_i \psi_i^T , \tag{20}$$

where $\Lambda$ is the diagonal matrix with ordered eigenvalues and $\Psi = (\psi_1, \psi_2, ..., \psi_n)$ is the matrix with columns corresponding to the ordered eigenvectors. The heat equation associated with the graph Laplacian, written in terms of the heat kernel $h_t$ and time $t$, is defined as

$$\frac{\partial h_t}{\partial t} = -\widehat{L} h_t .$$

Finally, the heat-kernel distance between nodes $u$ and $v$ at time $t$ is then computed as:

$$h_t(u, v) = \sum_{i=1}^{n} \exp -\lambda_i t \psi_i(u) \psi_i(v) . \tag{21}$$

**Definition B.4** (Resistance Distance (Randić & Klein, 1993)). Given a graph $G$ with $n$ nodes, the resistance distance between nodes $u$ and $v$ is given as:

$$\Omega_{u,v} = \Lambda_{i,i} + \Lambda_{j,j} - \Lambda_{i,j} - \Lambda_{j,i} , \tag{22}$$

where

$$\Lambda = \left( L + \frac{1}{n} \mathbf{1}_n \right)^+ ,$$

with $+$ denoting the Moore-Penrose pseudoinverse.

**Definition B.5** (Shortest-Path Distance). For a graph $G = (V, E)$, we define the shortest-path distance between two nodes $u, v \in V$ as the minimum number of edges in any path between $u$ and $v$.

### B.2.2. FEATURE METRICS

**Definition B.6** (Euclidean Distance $[d_F]$). Given our feature space $X \subset \mathbb{R}^k$, let $x, y \in \mathbb{R}^k$ be two arbitrary feature vectors. Then the Euclidean distance (also known as the L2 norm) between $x$ and $y$ is defined as

$$d_F(x, y) = \sqrt{\sum_{i=1}^{k} (x_i - y_i)^2} . \tag{23}$$

**Definition B.7** (Cosine Distance). Given a feature space $X \subset \mathbb{R}^k$, let $x, y \in \mathbb{R}^k$ be two arbitrary feature vectors. Then the cosine distance between $x$ and $y$ is defined as

$$d(x, y) = 1 - \frac{x \cdot y}{\|x\| \|y\|} , \tag{24}$$

where $x \cdot y$ is the dot product of $x$ and $y$, and $\|x\|$ is the L2 norm of $x$.

# C. Extended Dataset Descriptions

As we seek to evaluate the datasets in their capacity as graph-learning benchmarks, the formulation and properties of these datasets are worth detailed discussion. We summarize the basic statistics of these datasets in Table 3 and now describe the semantics and special characteristics of each dataset in turn.

## C.1. Social Sciences

### C.1.1. PROFESSIONAL COLLABORATIONS

In professional collaboration networks, nodes represent individuals and edges connect those with a direct working relationship.

*COLLAB* (Yanardag & Vishwanathan, 2015) is a dataset of physics collaboration networks, where nodes represent physics researchers and edges connect co-authors. Graphs are constructed as ego-networks around a central physics researcher, whose subfield (*high energy physics*, *condensed matter physics*, or *astrophysics*) acts as the classification for the graph as a whole. The data to build these graphs is drawn from three public collaboration datasets based on the arXiv (Leskovec et al., 2005).

Following a similar construction, the *IMDB* (Yanardag & Vishwanathan, 2015) datasets capture collaboration in movies. Graphs represent the ego-networks of actors and actresses, and edges connect those who appear in the same movie. The graph is classified according to the genre of the movies from which these shared acting credits are pulled. The task is to predict this genre, a binary classification between *Action* and *Comedy* in *IMDB-B*, and a multi-class classification between *Comedy*, *Romance*, and *Sci-Fi* in *IMDB-M*. (Note that genres are engineered to be mutually exclusive.) As one might expect, we note in Figure 3 the lower GNN performance for *IMDB-M* as compared to *IMDB-B*, suggesting that the multi-class graph classification is a more challenging task.

*COLLAB* and *IMDB* datasets share two notable features that distinguish them from the remaining datasets. First, graphs are constructed as ego-networks. This means that they have a diameter of at most two, in stark contrast to more structurally complex graphs from other datasets. This lends insight into why *COLLAB* and *IMDB* are classified as having low structural diversity (see Table 2). Second, node features are constructed from the node degree, rather than providing new information independent of graph structure. Thus, the low diversity scores with high variability that we see in Table 2 are simply a reflection of the node-degree distributions in these datasets.

### C.1.2. SOCIAL MEDIA

We study two graph-learning datasets with origins in social media: *REDDIT-B* and *REDDIT-M* (Yanardag & Vishwanathan, 2015).

In the these datasets, a graph models a thread lifted from a subreddit (i.e., a specific discussion community). Nodes represent Reddit users and edges connect users if at least one of them has replied to a comment made by the other (i.e., the users have had at least one interaction). Like *COLLAB* and *IMDB*, node features are constructed from the node degree. However, the *REDDIT* datasets score better on both structural and feature diversity (see Table 2). *REDDIT*'s higher structural diversity reflects the fact that its graphs are not ego-networks. Its more complex graph structures also account for the higher diversity in node features (i.e., node degrees).

The *REDDIT-B* dataset draws from four popular subreddits, two of which follow a Q&A structure (*IAmA*, *AskReddit*) and two of which are discussion-based (*TrollXChromosomes*, *atheism*). The task is to classify graphs based on these two styles of thread. *REDDIT-M* graphs are drawn from five subreddits, which also act as the graph class (*worldnews*, *videos*, *AdviceAnimals*, *aww* and *mildlyinteresting*).

Similar to the difference between the *IMDB* binary and multi-class datasets, we observe better GNN performance on the *REDDIT-B* benchmark as compared to *REDDIT-M* (see Figure 3), despite the fact that *REDDIT-M* boasts more than twice as many graphs (see Table 3).

## C.2. Life Sciences

### C.2.1. MOLECULES AND COMPOUNDS

Graphs provide an elegant way to model molecules, with nodes representing atoms and edges representing the chemical bonds between them.

The *AIDS* dataset (Riesen & Bunke, 2008), for example, is built from a repository of molecules in the AIDS Antiviral Screen Database of Active Compounds (NIH National Cancer Institute, 2004). Nodes have four features (Morris et al., 2020), most notably their unique atomic number. This is a binary-classification dataset in which molecules are classified by whether or not they demonstrate activity against the HIV virus. Among the 2 000 graphs in this dataset, note that the two classes are imbalanced: there are four times as many inactive molecules (1 600) as active (400). The *Mol-HIV (ogbg-molhiv)* dataset, introduced by the Open Graph Benchmark (Hu et al., 2020) with over 40,000 molecules from MoleculeNet (Wu et al., 2018b), proposes a similar task. Like *AIDS*, *MolHIV* classifies molecules into two classes based on their ability or inability to suppress the

| Dataset | $N$ | $\mu$ | $n$ $\sigma$ | $\mu$ | $m$ $\sigma$ | $\mu$ | $\delta$ $\sigma$ | $\mu$ | $\rho$ $\sigma$ | $\mu$ | $\gamma_1$ $\sigma$ | $\mu$ | $\gamma_{10}$ $\sigma$ |
|---|---|---|---|---|---|---|---|---|---|---|---|---|---|
| AIDS | 2000 | 15.69 | 13.69 | 32.39 | 30.02 | 2.01 | 0.20 | 0.19 | 0.08 | 0.29 | 0.08 | 0.28 | 0.06 |
| COLLAB | 5000 | 74.49 | 62.31 | 4914.43 | 12879.12 | 37.37 | 43.97 | 0.51 | 0.30 | 0.40 | 0.27 | 0.64 | 0.21 |
| DD | 1178 | 284.32 | 272.12 | 1431.32 | 1388.40 | 4.98 | 0.59 | 0.03 | 0.02 | 0.33 | 0.04 | 0.80 | 0.06 |
| Enzymes | 600 | 32.63 | 15.29 | 124.27 | 51.04 | 3.86 | 0.49 | 0.16 | 0.11 | 0.49 | 0.07 | 0.26 | 0.05 |
| IMDB-B | 1000 | 19.77 | 10.06 | 193.06 | 211.31 | 8.89 | 5.05 | 0.52 | 0.24 | 0.49 | 0.23 | 0.53 | 0.20 |
| IMDB-M | 1500 | 13.00 | 8.53 | 131.87 | 221.63 | 8.10 | 4.82 | 0.77 | 0.26 | 0.73 | 0.30 | 0.75 | 0.28 |
| MUTAG | 188 | 17.93 | 4.59 | 39.59 | 11.40 | 2.19 | 0.11 | 0.14 | 0.04 | 0.51 | 0.07 | 0.48 | 0.02 |
| MolHIV | 41127 | 25.51 | 12.11 | 54.94 | 26.43 | 2.14 | 0.11 | 0.10 | 0.04 | 0.42 | 0.11 | 0.33 | 0.08 |
| NCI1 | 4110 | 29.87 | 13.57 | 64.60 | 29.87 | 2.16 | 0.11 | 0.09 | 0.04 | 0.54 | 0.05 | 0.51 | 0.03 |
| Peptides | 10873 | 151.54 | 84.10 | 308.54 | 171.99 | 2.03 | 0.04 | 0.02 | 0.02 | 0.38 | 0.09 | 0.26 | 0.04 |
| Proteins | 1113 | 39.06 | 45.78 | 145.63 | 169.27 | 3.73 | 0.42 | 0.21 | 0.20 | 0.45 | 0.04 | 0.26 | 0.06 |
| Reddit-B | 2000 | 429.63 | 554.20 | 995.51 | 1246.29 | 2.34 | 0.31 | 0.02 | 0.03 | 0.47 | 0.03 | 0.45 | 0.08 |
| Reddit-M | 4999 | 508.52 | 452.62 | 1189.75 | 1133.65 | 2.25 | 0.20 | 0.01 | 0.01 | 0.48 | 0.02 | 0.43 | 0.07 |

Table 3: **Basic statistics of our** 13 **evaluated graph-learning datasets.** We show the number of graphs ($N$) as well as the mean and standard deviation of the number of nodes ($n$), the number of edges ($m$), the degree ($\delta$), the density ($\rho$), and mode complementarity at $t \in \{1, 10\}$ ($\gamma_1$, $\gamma_{10}$).

HIV virus. However, nodes in the *MolHIV* dataset have 9 features (compared to 4 in *AIDS*), including atomic number, chirality, and formal charge.

*MUTAG* (Debnath et al., 1991) is a relatively small molecular dataset compared with its peers, having only 188 graphs (Morris et al., 2020). However, it does have a similar number of node features (7). Graphs represent aromatic or heteroaromatic nitro compounds, which are divided into two classes by their mutagenicity towards the bacterium *S. typhimurium*, i.e. their ability to cause mutations its DNA. While small, the dataset contains an "extremely broad range of molecular structures" (Debnath et al., 1991), which we see reflected in a decent structural-diversity score in Table 2.

When introducing the dataset, Debnath et al. (1991) found that both specific sets of atoms and certain structural features (namely the presence of 3 or more fused rings), tended to be fairly well correlated with increased mutagenicity. This biologically supports that both modes could be informative, which is generally encouraging for graph-learning datasets. Thus, although we do not observe the desired performance separability in Figure 3, we cannot rule out that *MUTAG* could be a good benchmark, given a different task.

*NCI1* (Wale & Karypis, 2006) is a binary-classification dataset that splits molecules into two classes based on whether or not they have an inhibitary effect on the growth of human lung-cancer cell lines. Nodes have 37 features, a significant step up from other molecular datasets, which is reflected in the richness of *NCI1*'s feature diversity (see Table 2).

### C.2.2. LARGER CHEMICAL STRUCTURES

The *Peptides (Peptides-func)* dataset (Dwivedi et al., 2022) classifies peptides by their function. Each graph represents a peptide (i.e., a 1D amino acid chain), with nodes representing heavy (non-hydrogen) atoms and edges representing the bonds between them. *Peptides* is a multi-label-classification dataset, meaning that each peptide may belong to more than one of the 10 function classes (*Antibacterial*, *Antiviral*, *cell-cell communication*, etc.), with the average being 1.65 classes per peptide. Given the complexity of this task, it is perhaps unsurprising to notice that our tuned GNNs achieve the lowest AURUC on this dataset (see Figure 3).

*DD*, *ENZYMES*, and *Proteins (Proteins-full)* are all protein datasets. Each dataset draws upon the framework introduced by (Borgwardt et al., 2005) to model 3D structures of folded proteins as graphs, with nodes representing amino acids and edges connecting amino acids within a given proximity measured in Angstroms. Both the *Proteins-full* database introduced by (Borgwardt et al., 2005) and the *DD* database used by (Shervashidze et al., 2011) are based on data from Dobson & Doig (2003), which classifies 1 178 proteins as either enzymes (59%) or non-enzymes (41%). Thus, these two datasets are binary-classification datasets.

*Proteins-full* only uses 1 128 of the 1 178 proteins from Dobson & Doig (2003) but encodes 29 descriptive node features, whereas *DD* encodes none (Kersting et al., 2016; Morris et al., 2020). This distinction is responsible for Proteins-full's drastically higher feature diversity score and standard deviation, as seen in Table 2.

*ENZYMES* (Borgwardt et al., 2005) introduces a multi-class classification task on enzymes. Graphs represent enzymes from the BRENDA database (Schomburg et al., 2004),

| Aspect | Details |
|---|---|
| **Architectures** | $\{$GAT, GCN, GIN$\}$ |
| **Mode Perturbations** | $\{\varphi_o, \varphi_{eg}, \varphi_{cg}, \varphi_{cf}, \varphi_{rf}, \varphi_{rg}\}$ |
| **Tuning Strategy** | 5-fold CV $\times$ 64 consistent $\theta$ for each $(\varphi(D), \mathcal{A})$ |
| **Selection Metric** | Validation AUROC |
| **Evaluation Strategy** | Tuned model $(\varphi(D), \mathcal{A}, \theta)$ re-trained on distinct CV splits and random seeds, then evaluated on $\varphi(D)$'s test set. |

Table 4: **GNN setup to evaluate performance separability.** Parameter combinations are specified in Table 5. $\varphi_{rg}(\mathcal{D})$ perturbations were trained using a $\mathcal{R}_{F,1}(\cdot, 10)$ randomization. The top performing parameter combination $\theta^\star$ was selected based on validation statistics: primarily based on AUROC, with ties broken by accuracy and then loss.

which are classified according to the 6 Enzyme Commission top-level enzyme classes (EC classes). The dataset is roughly half the size of *DD* and *Proteins-full*, with only 600 enzymes (100 from each class). There are 18 node features, producing the high feature diversity observed in Table 2.

### C.3. Further Information

Several efforts have been made to consolidate information about the aforementioned datasets and their peers. In particular, we direct the interested reader to the TUDataset documentation (Morris et al., 2020), TU Dortmund's Benchmark Data Sets for Graph Kernels (Kersting et al., 2016), and PyTorch Geometric (Fey & Lenssen, 2019).

## D. Extended Experiments

### D.1. Extended Setup

With the RINGS framework, we introduce principled perturbations to graph learning datasets that allow us to evaluate the extent to which graph structure and node features are leveraged in a given task (*performance separability*) and to what extent the information in these modes is complementary or redundant (*mode complementarity*). We do so with a series of experiments, which we expand upon below.

### D.2. Extended Performance Separability

We begin with a note on the chosen GNN architectures, before examining and supplementing the results of our performance separability evaluations.

#### D.2.1. A NOTE ON GNN ARCHITECTURES

By far the most computationally expensive part of our experiments is tuning GNNs. While variation in the GNN models is desirable to ensure that our results are not overly

| Parameter | Values |
|---|---|
| Activation | `ReLU` |
| Batch Size | $\{64, 128\}$ |
| Dropout | $\{0.1, 0.5\}$ |
| Fold | $\{0, 1, 2, 3, 4\}$ |
| Hidden Dim | $\{128, 256\}$ |
| Learning Rate (LR) | $\{0.01, 0.001\}$ |
| Max Epochs | 200 |
| Normalization | `Batch` |
| Num Layers | 3 |
| Optimizer | `Adam` |
| Readout | `Sum` |
| Seed | $\{0, 42\}$ |
| Weight Decay | $\{0.0005, 0.005\}$ |

Table 5: **GNN tuning parameters.** For consistency, all $(\varphi(\mathcal{D}), \mathcal{A})$ were tuned across a consistent hyperparameter grid search.

dependent on a specific architecture, we were limited in the number of GNN models that we could train and evaluate over all of our datasets and their perturbations. As our goal is to evaluate the intrinsic quality of datasets as graph-learning benchmarks, we are more concerned with consistency across experiments than with using the newest methods, especially given finite computational resources.

In accordance with other "evaluation of evaluations" studies (Bechler-Speicher et al., 2024; Li et al., 2024), we opted to use several of the most common GNN architectures, namely GCN (Kipf & Welling, 2017), GAT (Veličković et al., 2018), and GIN (Xu et al., 2019). As noted by Li et al. (2024), GCN and GIN provide a good contrast in methodology, with GCN taking a spectral approach and GIN taking a spatial approach.

For each (dataset, perturbation, architecture) triple, we tune the GNN hyperparameters as outlined in Table 4, using the tuning parameters ($\theta$) stated in Table 5. The mean and standard deviation of accuracy and AUROC for our tuned models are tabulated in Table 6 and Table 7, respectively.

#### D.2.2. PERFORMANCE-SEPARABILITY RESULTS

Reasoning from first principles, we assert that the most instructive graph-learning datasets should have a separable original mode. In other words, we would like both the features and the graph structure to be informative and necessary to solve a given task. When a perturbed mode either outperforms or is non-separable from the original, then we can say that the perturbed mode is non-essential for the task at hand.

Consider the *Accuracy* and *AUROC* columns of Table 1. (Overall we find *AUROC* to be more expressive in determin-

| Dataset | | $\mu$ o | $\sigma$ o | $\mu$ cg | $\sigma$ cg | $\mu$ eg | $\sigma$ eg | $\mu$ rg | $\sigma$ rg | $\mu$ cf | $\sigma$ cf | $\mu$ rf | $\sigma$ rf |
|---|---|---|---|---|---|---|---|---|---|---|---|---|---|
| AIDS | GAT | 0.980 | 0.006 | 0.968 | 0.017 | 0.977 | 0.007 | 0.968 | 0.011 | 0.998 | 0.002 | 0.320 | 0.075 |
| | GCN | 0.982 | 0.005 | 0.950 | 0.016 | 0.983 | 0.004 | 0.837 | 0.058 | 0.997 | 0.003 | 0.340 | 0.051 |
| | GIN | 0.986 | 0.004 | 0.996 | 0.003 | 0.981 | 0.005 | 0.991 | 0.004 | 0.998 | 0.003 | 0.597 | 0.052 |
| COLLAB | GAT | 0.821 | 0.009 | 0.798 | 0.008 | 0.753 | 0.013 | 0.797 | 0.009 | 0.675 | 0.019 | 0.647 | 0.013 |
| | GCN | 0.827 | 0.009 | 0.798 | 0.008 | 0.753 | 0.014 | 0.790 | 0.009 | 0.705 | 0.018 | 0.712 | 0.012 |
| | GIN | 0.797 | 0.021 | 0.768 | 0.028 | 0.743 | 0.015 | 0.769 | 0.022 | 0.712 | 0.018 | 0.696 | 0.021 |
| DD | GAT | 0.580 | 0.085 | 0.747 | 0.029 | 0.598 | 0.086 | 0.696 | 0.090 | 0.721 | 0.045 | 0.579 | 0.049 |
| | GCN | 0.691 | 0.028 | 0.746 | 0.032 | 0.678 | 0.057 | 0.767 | 0.025 | 0.731 | 0.022 | 0.606 | 0.014 |
| | GIN | 0.683 | 0.037 | 0.601 | 0.073 | 0.670 | 0.050 | 0.594 | 0.074 | 0.710 | 0.030 | 0.597 | 0.018 |
| Enzymes | GAT | 0.470 | 0.054 | 0.415 | 0.065 | 0.530 | 0.044 | 0.406 | 0.053 | 0.414 | 0.044 | 0.173 | 0.032 |
| | GCN | 0.563 | 0.036 | 0.566 | 0.033 | 0.615 | 0.029 | 0.448 | 0.041 | 0.410 | 0.038 | 0.169 | 0.035 |
| | GIN | 0.572 | 0.058 | 0.332 | 0.068 | 0.607 | 0.094 | 0.338 | 0.062 | 0.400 | 0.048 | 0.182 | 0.029 |
| IMDB-B | GAT | 0.719 | 0.027 | 0.725 | 0.042 | 0.712 | 0.024 | 0.699 | 0.029 | 0.649 | 0.035 | 0.540 | 0.032 |
| | GCN | 0.732 | 0.033 | 0.719 | 0.039 | 0.716 | 0.022 | 0.704 | 0.025 | 0.692 | 0.024 | 0.606 | 0.034 |
| | GIN | 0.724 | 0.034 | 0.706 | 0.034 | 0.704 | 0.025 | 0.710 | 0.029 | 0.688 | 0.034 | 0.695 | 0.045 |
| IMDB-M | GAT | 0.493 | 0.023 | 0.489 | 0.017 | 0.478 | 0.028 | 0.489 | 0.019 | 0.431 | 0.029 | 0.349 | 0.026 |
| | GCN | 0.504 | 0.021 | 0.484 | 0.018 | 0.496 | 0.015 | 0.487 | 0.018 | 0.451 | 0.026 | 0.383 | 0.021 |
| | GIN | 0.475 | 0.047 | 0.476 | 0.032 | 0.484 | 0.025 | 0.489 | 0.019 | 0.452 | 0.031 | 0.398 | 0.041 |
| MUTAG | GAT | 0.707 | 0.051 | 0.677 | 0.083 | 0.667 | 0.042 | 0.688 | 0.038 | 0.755 | 0.087 | 0.624 | 0.079 |
| | GCN | 0.718 | 0.044 | 0.710 | 0.081 | 0.733 | 0.061 | 0.756 | 0.061 | 0.842 | 0.075 | 0.625 | 0.059 |
| | GIN | 0.767 | 0.140 | 0.814 | 0.068 | 0.707 | 0.056 | 0.801 | 0.068 | 0.821 | 0.078 | 0.689 | 0.042 |
| MolHIV | GAT | 0.962 | 0.003 | 0.653 | 0.151 | 0.964 | 0.001 | 0.955 | 0.005 | 0.966 | 0.001 | 0.965 | 0.001 |
| | GCN | 0.967 | 0.002 | 0.966 | 0.001 | 0.963 | 0.002 | 0.966 | 0.001 | 0.965 | 0.001 | 0.965 | 0.001 |
| | GIN | 0.968 | 0.002 | 0.965 | 0.001 | 0.963 | 0.001 | 0.965 | 0.002 | 0.965 | 0.001 | 0.965 | 0.001 |
| NCI1 | GAT | 0.562 | 0.056 | 0.580 | 0.037 | 0.506 | 0.004 | 0.575 | 0.031 | 0.666 | 0.014 | 0.500 | 0.001 |
| | GCN | 0.720 | 0.019 | 0.664 | 0.012 | 0.563 | 0.022 | 0.639 | 0.014 | 0.670 | 0.013 | 0.501 | 0.002 |
| | GIN | 0.769 | 0.016 | 0.681 | 0.030 | 0.561 | 0.019 | 0.648 | 0.011 | 0.667 | 0.015 | 0.548 | 0.039 |
| Peptides | GAT | 0.604 | 0.104 | 0.297 | 0.155 | 0.418 | 0.222 | 0.410 | 0.164 | 0.582 | 0.014 | 0.515 | 0.000 |
| | GCN | 0.671 | 0.006 | 0.673 | 0.008 | 0.651 | 0.007 | 0.633 | 0.013 | 0.610 | 0.010 | 0.515 | 0.000 |
| | GIN | 0.681 | 0.008 | 0.661 | 0.008 | 0.653 | 0.008 | 0.659 | 0.007 | 0.570 | 0.046 | 0.520 | 0.014 |
| Proteins | GAT | 0.600 | 0.015 | 0.620 | 0.038 | 0.599 | 0.007 | 0.614 | 0.034 | 0.723 | 0.024 | 0.610 | 0.015 |
| | GCN | 0.597 | 0.006 | 0.627 | 0.033 | 0.603 | 0.023 | 0.595 | 0.006 | 0.714 | 0.024 | 0.607 | 0.011 |
| | GIN | 0.603 | 0.011 | 0.601 | 0.049 | 0.601 | 0.012 | 0.622 | 0.046 | 0.701 | 0.034 | 0.606 | 0.011 |
| Reddit-B | GAT | 0.730 | 0.029 | 0.595 | 0.080 | 0.520 | 0.032 | 0.678 | 0.061 | 0.700 | 0.064 | 0.518 | 0.024 |
| | GCN | 0.781 | 0.018 | 0.742 | 0.026 | 0.540 | 0.050 | 0.794 | 0.023 | 0.823 | 0.019 | 0.580 | 0.039 |
| | GIN | 0.699 | 0.109 | | | 0.564 | 0.038 | 0.693 | 0.107 | 0.699 | 0.085 | 0.561 | 0.087 |
| Reddit-M | GAT | 0.422 | 0.021 | | | 0.331 | 0.060 | 0.307 | 0.097 | | | 0.284 | 0.039 |
| | GCN | 0.472 | 0.017 | | | 0.446 | 0.026 | 0.497 | 0.013 | | | 0.454 | 0.018 |
| | GIN | 0.474 | 0.027 | | | 0.420 | 0.041 | 0.422 | 0.051 | | | 0.501 | 0.042 |

Table 6: **Accuracy of tuned perturbed models underlying our performance-separability computations.** For Reddit-M, the complete-graph and complete-features perturbations failed to train due to memory problems. For Reddit-B, GIN failed for similar reasons.

ing separability, but we include both in our classifications.) Note that when training on $\varphi_{\text{eg}}$, we are removing any structural information. Comparing the performance of $\varphi_{\text{eg}}$ to that of the original, we can determine whether or not the structural information is being used at all. For example, for the *ENZYMES* dataset, we see that $eg > o$, which implies that the graph structure is unnecessary for solving the current classification task. Conversely, we have some tasks for which only the structural mode is essential. *MUTAG* is one example where $cf$ and $o$ are not separable, implying that node features are not essential and the task can be completed based on the graph structure alone. Given their dependency on a graph mode, we would consider this type of dataset

interesting for structural tasks, but we would prefer benchmarks that combine structural *and* feature information.

Moving on to the *Structure* and *Feature* columns of Table 1, we formalize the notion of whether or not each mode is informative for each dataset and associated task. Let us consider the structural perturbations $\{\Phi_{eg}, \Phi_{rg}, \Phi_{rg}\}$ and the performances of their tuned models. If the perturbed model performances are separably lower than the original, then we can classify the *structure* as informative (e.g., COLLAB). Similarly, turning to feature perturbations $\{\Phi_{cf}, \Phi_{rf}\}$ and their associated performances, if the perturbed model performances are separably lower than the original, then the *features* are considered informative (e.g. ENZYMES). In

| Dataset | | $\mu$ o | $\sigma$ o | $\mu$ cg | $\sigma$ cg | $\mu$ eg | $\sigma$ eg | $\mu$ rg | $\sigma$ rg | $\mu$ cf | $\sigma$ cf | $\mu$ rf | $\sigma$ rf |
|---|---|---|---|---|---|---|---|---|---|---|---|---|---|
| AIDS | GAT | 0.984 | 0.012 | 0.985 | 0.011 | 0.981 | 0.010 | 0.984 | 0.010 | 0.994 | 0.008 | 0.829 | 0.141 |
| | GCN | 0.986 | 0.010 | 0.954 | 0.020 | 0.986 | 0.009 | 0.982 | 0.011 | 0.995 | 0.006 | 0.874 | 0.055 |
| | GIN | 0.987 | 0.011 | 0.991 | 0.011 | 0.984 | 0.010 | 0.994 | 0.006 | 0.994 | 0.008 | 0.813 | 0.160 |
| COLLAB | GAT | 0.945 | 0.005 | 0.929 | 0.004 | 0.905 | 0.003 | 0.929 | 0.003 | 0.828 | 0.015 | 0.824 | 0.010 |
| | GCN | 0.949 | 0.004 | 0.929 | 0.003 | 0.905 | 0.003 | 0.926 | 0.003 | 0.847 | 0.013 | 0.877 | 0.008 |
| | GIN | 0.926 | 0.022 | 0.906 | 0.034 | 0.897 | 0.007 | 0.908 | 0.023 | 0.855 | 0.013 | 0.858 | 0.017 |
| DD | GAT | 0.642 | 0.163 | 0.801 | 0.029 | 0.720 | 0.198 | 0.766 | 0.126 | 0.773 | 0.030 | 0.703 | 0.113 |
| | GCN | 0.749 | 0.028 | 0.797 | 0.034 | 0.770 | 0.072 | 0.830 | 0.026 | 0.763 | 0.021 | 0.762 | 0.024 |
| | GIN | 0.762 | 0.053 | 0.755 | 0.120 | 0.769 | 0.044 | 0.733 | 0.130 | 0.751 | 0.038 | 0.747 | 0.059 |
| Enzymes | GAT | 0.767 | 0.033 | 0.729 | 0.044 | 0.797 | 0.022 | 0.729 | 0.033 | 0.730 | 0.028 | 0.513 | 0.032 |
| | GCN | 0.816 | 0.017 | 0.811 | 0.020 | 0.829 | 0.017 | 0.758 | 0.025 | 0.718 | 0.025 | 0.505 | 0.030 |
| | GIN | 0.828 | 0.032 | 0.671 | 0.047 | 0.840 | 0.053 | 0.682 | 0.044 | 0.722 | 0.033 | 0.534 | 0.031 |
| IMDB-B | GAT | 0.811 | 0.030 | 0.822 | 0.036 | 0.803 | 0.025 | 0.791 | 0.026 | 0.694 | 0.035 | 0.561 | 0.037 |
| | GCN | 0.837 | 0.028 | 0.821 | 0.037 | 0.803 | 0.023 | 0.793 | 0.026 | 0.741 | 0.037 | 0.644 | 0.042 |
| | GIN | 0.811 | 0.040 | 0.795 | 0.039 | 0.789 | 0.026 | 0.798 | 0.030 | 0.742 | 0.036 | 0.759 | 0.053 |
| IMDB-M | GAT | 0.687 | 0.024 | 0.673 | 0.019 | 0.668 | 0.022 | 0.683 | 0.020 | 0.597 | 0.022 | 0.527 | 0.023 |
| | GCN | 0.704 | 0.020 | 0.668 | 0.019 | 0.684 | 0.021 | 0.679 | 0.020 | 0.623 | 0.029 | 0.566 | 0.019 |
| | GIN | 0.671 | 0.037 | 0.658 | 0.027 | 0.674 | 0.018 | 0.677 | 0.019 | 0.623 | 0.028 | 0.594 | 0.036 |
| MUTAG | GAT | 0.836 | 0.063 | 0.761 | 0.119 | 0.867 | 0.081 | 0.840 | 0.062 | 0.898 | 0.062 | 0.618 | 0.110 |
| | GCN | 0.846 | 0.048 | 0.744 | 0.075 | 0.814 | 0.068 | 0.825 | 0.058 | 0.908 | 0.066 | 0.585 | 0.092 |
| | GIN | 0.874 | 0.155 | 0.874 | 0.067 | 0.814 | 0.075 | 0.865 | 0.048 | 0.882 | 0.078 | 0.838 | 0.079 |
| MolHIV | GAT | 0.689 | 0.016 | 0.633 | 0.048 | 0.452 | 0.022 | 0.654 | 0.019 | 0.685 | 0.023 | 0.359 | 0.017 |
| | GCN | 0.726 | 0.014 | 0.630 | 0.011 | 0.588 | 0.007 | 0.645 | 0.016 | 0.691 | 0.028 | 0.361 | 0.018 |
| | GIN | 0.777 | 0.013 | 0.696 | 0.022 | 0.590 | 0.019 | 0.646 | 0.023 | 0.670 | 0.028 | 0.409 | 0.028 |
| NCI1 | GAT | 0.572 | 0.135 | 0.680 | 0.051 | 0.679 | 0.017 | 0.664 | 0.065 | 0.727 | 0.016 | 0.683 | 0.016 |
| | GCN | 0.797 | 0.017 | 0.735 | 0.013 | 0.647 | 0.029 | 0.685 | 0.017 | 0.731 | 0.017 | 0.648 | 0.022 |
| | GIN | 0.843 | 0.014 | 0.741 | 0.031 | 0.640 | 0.034 | 0.701 | 0.022 | 0.722 | 0.016 | 0.656 | 0.024 |
| Peptides | GAT | 0.553 | 0.042 | 0.401 | 0.069 | 0.444 | 0.093 | 0.446 | 0.065 | 0.548 | 0.006 | 0.401 | 0.006 |
| | GCN | 0.582 | 0.004 | 0.596 | 0.006 | 0.561 | 0.008 | 0.580 | 0.006 | 0.549 | 0.008 | 0.400 | 0.007 |
| | GIN | 0.591 | 0.005 | 0.571 | 0.012 | 0.565 | 0.004 | 0.573 | 0.007 | 0.536 | 0.008 | 0.447 | 0.027 |
| Proteins | GAT | 0.775 | 0.018 | 0.752 | 0.042 | 0.786 | 0.016 | 0.764 | 0.035 | 0.758 | 0.022 | 0.720 | 0.030 |
| | GCN | 0.782 | 0.021 | 0.782 | 0.027 | 0.773 | 0.028 | 0.780 | 0.024 | 0.758 | 0.027 | 0.730 | 0.031 |
| | GIN | 0.782 | 0.028 | 0.735 | 0.092 | 0.736 | 0.061 | 0.751 | 0.046 | 0.742 | 0.034 | 0.765 | 0.032 |
| Reddit-B | GAT | 0.798 | 0.022 | 0.763 | 0.049 | 0.749 | 0.148 | 0.740 | 0.088 | 0.721 | 0.082 | 0.624 | 0.178 |
| | GCN | 0.846 | 0.014 | 0.817 | 0.019 | 0.639 | 0.195 | 0.875 | 0.016 | 0.862 | 0.015 | 0.797 | 0.040 |
| | GIN | 0.748 | 0.066 | | | 0.734 | 0.097 | 0.783 | 0.051 | 0.727 | 0.059 | 0.773 | 0.039 |
| Reddit-M | GAT | 0.717 | 0.014 | | | 0.733 | 0.053 | 0.644 | 0.120 | | | 0.717 | 0.036 |
| | GCN | 0.766 | 0.009 | | | 0.759 | 0.013 | 0.805 | 0.006 | | | 0.777 | 0.009 |
| | GIN | 0.770 | 0.018 | | | 0.742 | 0.035 | 0.746 | 0.031 | | | 0.797 | 0.016 |

Table 7: **AUROC of tuned perturbed models underlying our performance-separability computations.** For Reddit-M, the complete-graph and complete-features perturbations failed to train due to memory problems. For Reddit-B, GIN failed for similar reasons.

the case that AUROC and accuracy yield different outcomes for whether or not a mode is informative then this is denoted as *(un)informative*.

For the judgment depicted in the *Evaluation* column of Table 1, we score datasets on a scale from 0 to 5 as $1.5 \cdot S + 1 \cdot F$, where $S, F \in \{0, 1, 2\}$, mapping "uninformative" to 0, "(un)informative" to 1, and informative to 2, and giving the structural mode higher weight than the feature mode because we are dealing with *graph*-learning datasets. The final evaluation results from binning the scores as

$$[0, 1] \mapsto --, \ (1, 2] \mapsto -,$$
$$(2, 3] \mapsto \circ, \ (3, 4] \mapsto +, \ (4, 5] \mapsto ++.$$

While we include this categorization to simplify the message

conveyed by our results, we recommend considering the full, uncondensed separability results when assessing the quality of individual datasets.

D.2.3. PERFORMANCE-SEPARABILITY EVALUATION

For each dataset, we evaluate the *performance separability* between its perturbed versions using two different statistics in our permutation tests (Kolmogorov-Smirnov and Wilcoxon rank-sum) as well as two different $\alpha$-levels as significance cutoffs (0.01 and 0.005). Notably, we see very few differences in the separability results and the associated classification of datasets, confirming the robustness of our approach.

| Component | Specifications |
|---|---|
| *Available CPUs* | Intel Xeon (Haswell, Broadwell, Skylake, Cascade Lake, Sapphire Rapids, Emerald Rapids) |
| | Intel Xeon (6134, 6248R, 6142M, 6128, 6136, E5620) |
| | Intel Platinum (8280L, 8468, 8562Y+) |
| | AMD Opteron (6164 HE, 6234, 6376 (x2), 6272, 6128) |
| | AMD EPYC (7742, 7713, 7413, 7262) |
| *Available GPUs* | NVIDIA Tesla (K80, P100, V100, A100, H100, H200) |
| | NVIDIA Quadro (RTX 8000, RTX 6000) |
| | AMD MI100 |

Table 8: **Summary of Compute Resources.** Especially when tuning our models used to generate the *performance separability* results as seen in Tables 6 and 7, we relied heavily on high performance cluster units at multiple institutions that supported our work with the above hardware specifications.

| Parameter | Values |
|---|---|
| Feature Metric | `Euclidean` |
| Perturbations | $\{\varphi_o, \varphi_{e*}, \varphi_{c*}, \varphi_{r*}\}$ |
| Structural Metric | `Diffusion` |
| Matrix Norm | $L_{1,1}$ |
| Diffusion Steps | $\{1, \cdots, 10\}$ |
| Seed* | $0, 2^1, 2^2, 2^3, 2^4$ |

Table 9: **Complementarity-calculation setup for randomized mode perturbations.** We compute complementarity under perturbations $\gamma_\varphi^{1,1}$ as specified in the above table, with $* \in \{f, g\}$. Without being subject to gradient descent in our calculations, we can also compute $\gamma_{ef}^{1,1}$. For a definition of `Diffusion`, see Definition B.2 and a note our treatment of disconnected graphs in Appendix B. Finally, seed is only varied over the random perturbations $\varphi_{r*}$ and our results are aggregated based on the average complementarity.

#### D.2.4. SUPPLEMENTARY EXTENSIBILITY EXPERIMENTS

The RINGS framework was developed from first principles to be a tool for the graph-learning community. In the spirit of inspiring and facilitating new and more detailed analyses of graph-learning datasets, RINGS was designed to be modular and configurable.

Just as adaptability to other feature and structure metrics has been established in Appendix B.2, we continue to explore the extensibility of RINGS in this section, demonstrating how it can accomodate other tasks (graph regression), architectures (transformer architectures), and a different granularity of results (graph-level performance results).

**Extending to graph-level performance results.** Inspired by reviewer feedback during the rebuttal phase, we present a preview of a deeper suite of analyses enabled by RINGS —specifically, a graph-level investigation of performance

that probes the separability of individual samples. Instead of asking how perturbations affect overall accuracy, we ask a more granular question: *How similar are the sets of graphs that two given mode perturbations classify correctly?* To explore this, we compute the average similarity of correctly classified graph sets across multiple test splits and random seeds. Table 12 outlines the configurations for two datasets from different categories in our dataset taxonomy: MUTAG and PROTEINS.

For each $(\varphi(\mathcal{D}), \mathcal{A})$ pair, we extract the set of correctly classified graphs and compare them across mode perturbations. We report two similarity metrics: the *Jaccard Similarity*, which measures the proportion of overlap between two sets, and an *Asymmetric Overlap*, which quantifies the fraction of correctly classified samples under a perturbation that are also classified correctly in the original setting. Formally, we define

$$\text{Jaccard}(A, A') = \frac{|A \cap A'|}{|A \cup A'|}, \quad \text{Asymm}(A', A) = \frac{|A \cap A'|}{|A'|},$$

where $A$ and $A'$ denote the sets of graphs correctly classified for by the two models being compared.

Our findings are visualized in Figure 8. Across both datasets, most configuration pairs show strong agreement in the graphs they classify correctly. However, in Figure 8b, the $\varphi_{cf}$ perturbation exhibits consistently lower similarity scores across all architectures, suggesting that it correctly classifies a distinct—and likely outlier—subset of graphs. Combined with the results in the main paper, which show high accuracy but low AUROC for Proteins under $\varphi_{cf}$, this suggests that these models are exploiting superficial structural cues to memorize the dominant class, rather than learning meaningful patterns.

By analyzing graph-level similarity across perturbation modes and architectures, we provide a more nuanced view of model behavior—one that moves beyond coarse aggregate metrics to capture variation at the level of individual

| Dataset | Accuracy | AUROC | Structure | Features | Evaluation |
|---|---|---|---|---|---|
| AIDS | cf > cg > rg > o > eg > rf | cf/cg/eg/o/rg > rf | uninformative | uninformative | −− |
| COLLAB | o > cg/rg > eg > cf/rf | o > cg/rg > eg > rf > cf | informative | informative | ++ |
| DD | rg > cg > cf > eg/o > rf | rg > cf/cg/eg/o/rf | uninformative | uninformative | −− |
| Enzymes | eg > cg/o > rg > cf > rf | eg/o > cg > rg > cf > rf | uninformative | informative | − |
| IMDB-B | cf/cg/eg/o/rf/rg | cg/o > eg/rg > cf/rf | uninformative | (un)informative | −− |
| IMDB-M | cg/eg/o/rg > cf > rf | o > eg/rg > cg > cf > rf | (un)informative | informative | + |
| MUTAG | cf/cg/eg/o/rg > rf | cf/cg/eg/o/rf/rg | uninformative | uninformative | −− |
| MolHIV | o > cf/cg/rg > rf > eg | o > cf/cg > rg > eg > rf | informative | informative | ++ |
| NCI1 | o > cg > cf > rg > eg/rf | o > cg > cf > rg > eg/rf | informative | informative | ++ |
| Peptides | o > cg > rg > eg > cf > rf | cg > o > rg > eg > cf > rf | (un)informative | informative | + |
| Proteins | cf > cg/eg/o/rf/rg | cf/cg/eg/o/rf/rg | uninformative | uninformative | −− |
| Reddit-B | cf > rg > o > cg > eg/rf | rg > cf > o > cg/eg/rf | uninformative | uninformative | −− |
| Reddit-M | rf/rg > o > eg | rg > rf > o > eg | uninformative | uninformative | −− |

Table 10: **Measuring *performance separability* between different versions of the same dataset: Wilcoxon rank-sum test.** Supplementing Table 1, we use the Wilcoxon rank-sum as a test statistic to replace the KS statistic in our permutation tests, with an otherwise identical setup (10 000 random permutations; $\alpha$-level of 0.01; Bonferroni correction within each dataset). The results are substantively the same, with the exceptions that IMDB-B gets downgraded from fair to poor, and MUTAG gets downgraded from poor to very poor.

| Dataset | Accuracy | AUROC | Structure | Features | Evaluation |
|---|---|---|---|---|---|
| AIDS | cf > cg > rg > eg/o > rf | cf/cg/rg > eg/o > rf | uninformative | uninformative | −− |
| COLLAB | o > cg/rg > eg > cf/rf | o > cg/rg > eg > rf > cf | informative | informative | ++ |
| DD | rg > cg > cf > eg/o > rf | rg > cf/cg/eg/o/rf | uninformative | uninformative | −− |
| Enzymes | eg > cg/o > rg > cf > rf | eg/o > cg > rg > cf > rf | uninformative | informative | − |
| IMDB-B | cf/cg/eg/o/rf/rg | o > cg > eg/rg > cf/rf | (un)informative | (un)informative | ○ |
| IMDB-M | cg/eg/o/rg > cf > rf | o > cg/eg/rg > cf > rf | (un)informative | informative | + |
| MUTAG | cf/cg/o/rg > eg > rf | cf/o > cg/eg/rf/rg | (un)informative | uninformative | − |
| MolHIV | o > cf/cg/rg > rf > eg | o > cf/cg > rg > eg > rf | informative | informative | ++ |
| NCI1 | o > cg > cf > rg > eg > rf | o > cg > cf > rg > eg/rf | informative | informative | ++ |
| Peptides | o > cg > rg > eg > cf > rf | cg > o > rg > eg > cf > rf | (un)informative | informative | + |
| Proteins | cf > cg/eg/o/rf/rg | cf/cg/eg/o/rf/rg | uninformative | uninformative | −− |
| Reddit-B | cf > o/rg > cg > eg/rf | rg > cf > o > cg/eg/rf | uninformative | uninformative | −− |
| Reddit-M | rf/rg > o > eg | rg > rf > o > eg | uninformative | uninformative | −− |

Table 11: **Measuring *performance separability* between different versions of the same dataset: $\alpha \leq 0.005$.** Supplementing Table 1, we test at an $\alpha$-level of 0.005 instead of 0.01, using an otherwise identical setup (10 000 random permutations; KS statistic; Bonferroni correction within each dataset). The results are substantively identical to Table 1.

samples. We hope that a deeper investigation of graph-level performance through RINGS will uncover opportunities for dataset improvement and inform the design of robustness-enhancing interventions.

**Extending to regression datasets.** While the primary analysis of the paper concerns graph *classification* datasets, RINGS is intrinsically task-agnostic. As a proof of concept, we also apply RINGS to two common graph regression datasets, QM9 and ZINC-12k. As recommended by their authors, we use MAE as the performance metric for both datasets.

*Dataset descriptions.* QM9 (Wu et al., 2018a) is a molecular dataset of 133 885 graphs. Its name originates from its focus on *quantum mechanical* properties and the makeup of the dataset: stable, organic molecules with *nine* or fewer heavy

atoms. The PyG version has eleven node features, the first five being a one-hot encoding of the atom type, and the rest representing measures such as the hybridization state and the number of hydrogen atoms. Nineteen regression tasks seek to predict a variety molecular properties. We focus on task 0 (as defined by PyG), regressing the dipole moment, which describes the spatial distribution of electrons.

The ZINC-12k dataset (Dwivedi et al., 2023), which contains 12 000 molecules, is a subset of the larger ZINC dataset (Sterling & Irwin, 2015), which contains 250 000 molecules. In ZINC-12k, each node has only one feature, which encodes the type of atom. The task is to regress constrained solubility, defined as the water-octanol partition coefficient ($logP$) minus the synthetic accessibility score ($SAS$) and the number of cycles with more than six atoms. Constrained solubility has implications in drug design, and thus has been

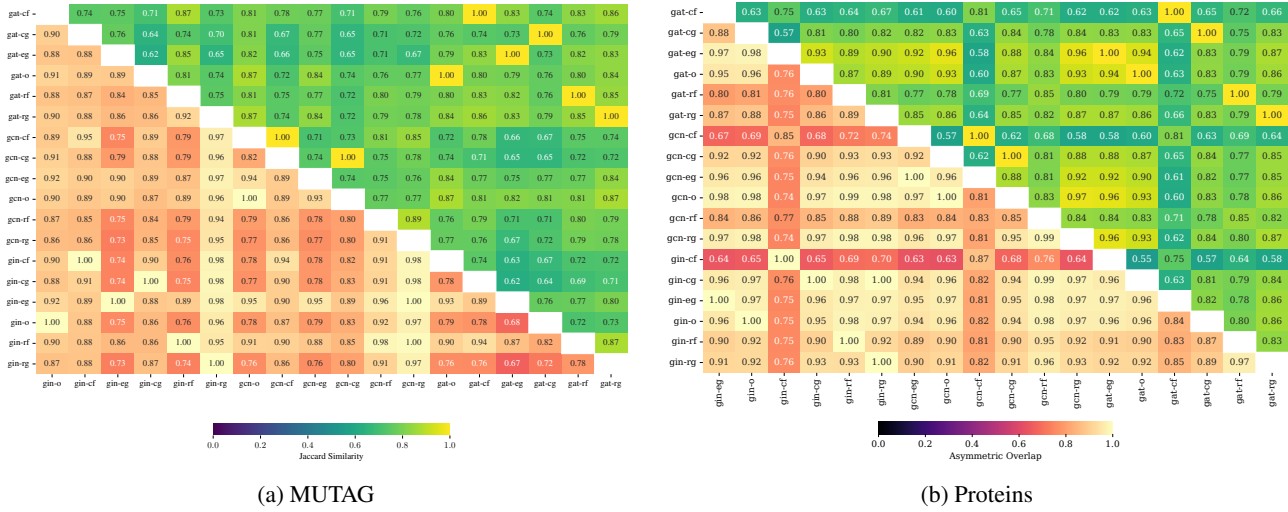

Figure 8: **Graph-level agreement between (mode, architecture) pairs.** Jaccard similarity (upper triangular) and asymmetric overlap (lower triangular) quantify the similarity between sets of graphs correctly classified by different models on the MUTAG (a) and Proteins (b) datasets. Higher values indicate a larger shared set of correctly classified samples between $(\varphi(\mathcal{D}), \mathcal{A})$ pairs.

| Aspect | Details |
|---|---|
| **Global (PyTorch) Seeds** | $\{42, 7, 123, 56, 89\}$ |
| **Test/Train Split Seeds** | $\{67, 23, 77, 88, 54\}$ |
| **Transform Seeds** | Random Modes: $\{34, 12, 99, 45, 10\}$ |
| | Fixed Modes: $\{34\}$ |
| **Evaluation Strategy** | Tuned model $(\varphi(D), \mathcal{A}, \theta)$ re-trained on distinct CV splits and all unique groupings of random seeds, then evaluated on the test set of $\varphi(D)$. |

Table 12: **Granular evaluation setup.** For our extended experiments—including graph-level analyses, regression tasks, and new architectures—we introduce additional controls to account for randomness in performance separability: (1) model initialization via the PyTorch global seed, (2) train/test splitting, and (3) perturbation generation (applicable to random-graph and random-features modes). To ensure consistency across datasets, we use the same three sets of five unique seeds.

| Mode | QM9 | ZINC-12k |
|---|---|---|
| cg | $0.39 \pm 0.05$ | $1.31 \pm 0.02$ |
| eg | $0.65 \pm 0.03$ | $0.98 \pm 0.03$ |
| cf | $\mathbf{0.07 \pm 0.01}$ | $1.40 \pm 0.02$ |
| o | $0.60 \pm 0.02$ | $\mathbf{0.80 \pm 0.03}$ |
| rf | $1.44 \pm 0.02$ | $1.48 \pm 0.01$ |
| rg | $0.93 \pm 0.01$ | $1.39 \pm 0.02$ |

Table 13: **Performance results for regression datasets using the GCN architecture.** For the QM9 and ZINC-12k regression datasets, each entry is reported as $\mu_{\text{MAE}} \pm \sigma_{\text{MAE}}$.

used as the target for molecular graph generation models, as in (Jin et al., 2018) and (Kusner et al., 2017).

*Performance separability results.* Using the fine-grained evaluation setup described in Table 12, we evaluate the performance of the two regression datasets using the GCN architecture, adapting the procedure to minimize MAE instead of maximizing AUROC. We report these results in Table 13. Note that we have far fewer successful runs as for our the graph classification experiments in the main paper (see Appendix D.2.4).

We thus opt to use bootstrapping, sampling $10\,000$ times with a confidence interval of $99\%$, yielding the results shown in Table 14. As RINGS was developed from first principles, we apply the same classification schema to the relevance of information in each mode and the overall dataset evaluation.

*Interpretation of results.* For QM9, performance-separability results suggest that neither the original mode nor its perturbations are particularly informative for the task. Notably, the original mode underperforms both $\varphi_{\text{cf}}$ and $\varphi_{\text{cg}}$, raising concerns about its suitability as a default configuration. Depending on the diversity of available modes, this may warrant realignment—or even deprecation—of QM9 in its current form. In contrast, ZINC-12k yields more encouraging results: The original mode consistently outperforms all perturbations, indicating that it captures task-relevant structure, and yielding a $++$ overall classification. Interestingly, graph perturbations tend to outperform feature perturbations, suggesting that node features play a more critical role in this

| Dataset | MAE | Structure | Features | Evaluation |
|---------|-----|-----------|----------|------------|
| QM9 | cf < cg < o < eg < rg < rf | uninformative | unformative | $--$ |
| ZINC-12k | o < eg < cg < rg/cf < rf | informative | informative | $++$ |

Table 14: **Performance separability for regression datasets following results in Table 13.** Separability is computed using bootstrapping, sampling $10,000$ times with a confidence interval of 99%. The evaluation is based on the performance separability schema outlined in Table 1.

task than the underlying graph structure. For both datasets, a more exhaustive training routine—as employed in the main paper—is necessary to draw definitive conclusions regarding their placement within the regression taxonomy.

**Extending to transformer architectures.** As recommended by our reviewers, we further demonstrate that RINGS also generalizes nicely to transformer architectures such as the GPS Transformer, introduced by Rampásek et al. (2022). In contrast to GCN, GIN and GAT, GPS combines local message passing with global attention to overcome the expressivity and scalability limits of traditional GNNs. We compare the performance separability when considering 4 architectures across three datasets (NCI1, MUTAG, Proteins). Note that other transformer architectures, such as the Graphformer architecture proposed by Yang et al. (2021) that we originally planned to include in our extended experiments, are tailored to scenarios with non-graph data, which lie beyond the scope of this work.

*Performance-separability results.* Accuracy and AUROC results for the GPS architecture are reported in Table 16 and Table 15, respectively, alongside newly generated results for GCN, GAT, and GIN—each run under the exact randomization schema outlined in Table 12.

Our goal is to assess whether incorporating a transformer-based architecture, specifically GPS, meaningfully alters performance separability outcomes. To this end, we compare separability results that include GPS with those based solely on the core architectures (GCN, GAT, GIN), as shown in Table 17. As a reminder, performance separability is computed using the best-performing architecture for each mode.

To assess statistical significance, we apply bootstrapping with 10,000 resamples and report 99% confidence intervals, as shown in Table 17. We then apply the RINGS classification schema to evaluate the informativeness of each mode and determine the overall dataset classification.

*Interpretation of results.* We find that the inclusion of GPS has negligible impact on performance separability and dataset classification. This validates our decision to focus on the core architectures in the main paper, which are computationally lighter, yet still sufficient to support the conclusions drawn about dataset behavior.

**Computational challenges.** Rigorous evaluation of GNNs and their benchmark datasets introduces significant computational challenges. Despite access to substantial computing resources, a notable portion of model trainings failed to converge—particularly under the more granular and stringent experimental setups introduced in this work. For example, even in our original experimental configurations, GIN models on REDDIT-B could not be successfully tuned or trained, and REDDIT-M experiments under the *complete-graph* and *complete-features* perturbations (see Figure 3) also failed due to memory limitations or convergence issues. Our extended evaluations for ZINC, QM9, GPS, and graph-level analyses faced similar constraints, resulting in smaller sample sizes for these experiments relative to those reported in the main results.

At the graph level, we achieved full coverage for MUTAG and 97% coverage for Proteins (based on the grids established in Table 12), but other datasets were hindered by incomplete runs. These issues underscore the need for more adaptive configuration and tuning strategies to ensure stable model convergence across diverse datasets and architectures. Understanding and mitigating these failure modes remains an important direction for future work. Our results reflect both the potential and the practical limitations of large-scale, rigorous GNN evaluation pipelines such as RINGS.

### D.3. Extended Mode Complementarity

In our mode-complementarity computations, we need to distinguish fixed modes (original, empty, complete) from randomized modes (random, shuffled). To compute mode complementarity for the randomized modes, we use five different randomization seeds for each graph, resulting in the setup summarized in Appendix D.2.2.

To study how perturbations impact the complementarity of a dataset, we refer to Figure 4 (see Figure 9 for a supplementary visualization of mode complementarities computed for 7 OGB datasets). In the top panel, we compare the original complementarity score using the $L_{1,1}$ norm, $\gamma^{1,1}$, to the complementarity under fixed perturbations $\gamma_\varphi^{1,1}$, where $\varphi \in \{\varphi_{\mathrm{cg}}, \varphi_{\mathrm{eg}}, \varphi_{\mathrm{cf}}, \varphi_{\mathrm{ef}}\}$, with the randomized perturbations shown in the bottom panel. Note the symmetry introduced by Theorem 2.15, i.e., that $\gamma_{e*}^{1,1} = 1 - \gamma_{c*}^{1,1}$. These perturbations, as per Proposition A.10, effectively measure

| Dataset | | $\mu$ o | $\sigma$ o | $\mu$ cg | $\sigma$ cg | $\mu$ eg | $\sigma$ eg | $\mu$ rg | $\sigma$ rg | $\mu$ cf | $\sigma$ cf | $\mu$ rf | $\sigma$ rf |
|---|---|---|---|---|---|---|---|---|---|---|---|---|---|
| MUTAG | GAT | 0.882 | 0.100 | 0.782 | 0.163 | 0.877 | 0.123 | 0.867 | 0.104 | 0.920 | 0.050 | 0.658 | 0.143 |
| | GCN | 0.885 | 0.097 | 0.827 | 0.140 | 0.862 | 0.097 | 0.851 | 0.091 | 0.903 | 0.062 | 0.598 | 0.136 |
| | GIN | 0.926 | 0.072 | 0.887 | 0.162 | 0.844 | 0.107 | 0.913 | 0.058 | 0.914 | 0.054 | 0.846 | 0.095 |
| | GPS | 0.727 | 0.245 | 0.483 | 0.399 | 0.415 | 0.386 | 0.476 | 0.398 | 0.719 | 0.330 | 0.849 | 0.180 |
| NCI1 | GAT | 0.552 | 0.140 | 0.672 | 0.056 | 0.667 | 0.026 | 0.655 | 0.070 | 0.738 | 0.017 | 0.683 | 0.033 |
| | GCN | 0.794 | 0.020 | 0.736 | 0.015 | 0.646 | 0.027 | 0.685 | 0.032 | 0.745 | 0.014 | 0.648 | 0.038 |
| | GIN | 0.844 | 0.017 | 0.740 | 0.046 | 0.628 | 0.041 | 0.693 | 0.036 | 0.731 | 0.019 | 0.653 | 0.032 |
| | GPS | 0.661 | 0.097 | 0.626 | 0.018 | 0.677 | 0.064 | 0.677 | 0.058 | 0.675 | 0.035 | 0.671 | 0.047 |
| PROTEINS | GAT | 0.785 | 0.053 | 0.759 | 0.048 | 0.791 | 0.051 | 0.769 | 0.057 | 0.769 | 0.030 | 0.733 | 0.040 |
| | GCN | 0.787 | 0.058 | 0.787 | 0.047 | 0.777 | 0.059 | 0.786 | 0.045 | 0.762 | 0.032 | 0.750 | 0.039 |
| | GIN | 0.788 | 0.045 | 0.769 | 0.035 | 0.780 | 0.050 | 0.769 | 0.036 | 0.731 | 0.051 | 0.770 | 0.042 |
| | GPS | 0.738 | 0.086 | 0.608 | 0.171 | 0.754 | 0.040 | 0.602 | 0.163 | 0.715 | 0.102 | 0.766 | 0.034 |

Table 15: **AUROC results for GPS and core architectures.** For each architecture, we report each mode's mean AUROC and its standard deviation. We do this for a subset of our datasets, namely Proteins, NCI1, and MUTAG.

| Dataset | | $\mu$ o | $\sigma$ o | $\mu$ cg | $\sigma$ cg | $\mu$ eg | $\sigma$ eg | $\mu$ rg | $\sigma$ rg | $\mu$ cf | $\sigma$ cf | $\mu$ rf | $\sigma$ rf |
|---|---|---|---|---|---|---|---|---|---|---|---|---|---|
| MUTAG | GAT | 0.736 | 0.102 | 0.701 | 0.099 | 0.654 | 0.106 | 0.677 | 0.079 | 0.736 | 0.104 | 0.647 | 0.104 |
| | GCN | 0.744 | 0.099 | 0.783 | 0.107 | 0.736 | 0.106 | 0.789 | 0.080 | 0.834 | 0.077 | 0.631 | 0.095 |
| | GIN | 0.853 | 0.087 | 0.826 | 0.137 | 0.701 | 0.125 | 0.845 | 0.077 | 0.839 | 0.067 | 0.682 | 0.082 |
| | GPS | 0.570 | 0.176 | 0.485 | 0.185 | 0.465 | 0.179 | 0.485 | 0.180 | 0.590 | 0.159 | 0.648 | 0.105 |
| NCI1 | GAT | 0.550 | 0.055 | 0.578 | 0.043 | 0.499 | 0.030 | 0.565 | 0.033 | 0.680 | 0.018 | 0.496 | 0.030 |
| | GCN | 0.717 | 0.025 | 0.669 | 0.018 | 0.538 | 0.021 | 0.640 | 0.027 | 0.684 | 0.015 | 0.499 | 0.029 |
| | GIN | 0.769 | 0.020 | 0.690 | 0.026 | 0.541 | 0.022 | 0.647 | 0.031 | 0.675 | 0.024 | 0.541 | 0.041 |
| | GPS | 0.492 | 0.034 | 0.528 | 0.016 | 0.489 | 0.038 | 0.497 | 0.039 | 0.610 | 0.033 | 0.494 | 0.028 |
| PROTEINS | GAT | 0.612 | 0.052 | 0.631 | 0.055 | 0.601 | 0.039 | 0.608 | 0.049 | 0.722 | 0.038 | 0.608 | 0.034 |
| | GCN | 0.593 | 0.033 | 0.627 | 0.069 | 0.600 | 0.046 | 0.593 | 0.029 | 0.725 | 0.036 | 0.610 | 0.028 |
| | GIN | 0.609 | 0.033 | 0.602 | 0.043 | 0.602 | 0.038 | 0.607 | 0.050 | 0.705 | 0.039 | 0.606 | 0.028 |
| | GPS | 0.582 | 0.049 | 0.547 | 0.083 | 0.590 | 0.028 | 0.532 | 0.068 | 0.669 | 0.109 | 0.591 | 0.027 |

Table 16: **Accuracy results for GPS and core architectures.** For each architecture, we report each mode's mean accuracy and its standard deviation. We do this for a subset of our datasets, namely Proteins, NCI1, and MUTAG.

| Dataset | GPS Results | Accuracy | AUROC | Structure | Features | Evaluation |
|---|---|---|---|---|---|---|
| MUTAG | Present | o/rg/cf/cg > eg > rf | o/cf/rg/cg/eg/rf | uninformative | uninformative | −− |
| | Absent | o/rg/cf/cg > eg > rf | o/cf/rg/cg/eg > rf | uninformative | uninformative | −− |
| NCI1 | Present | o > cg/cf > rg > eg/rf | o > cf/cg > rg > rf/eg | informative | informative | ++ |
| | Absent | o > cg/cf > rg > eg/rf | o > cf/cg > rg > rf > eg | informative | informative | ++ |
| Proteins | Present | cf > cg > o/rf/rg/eg | eg/o/cg/rg > rf/cf | uninformative | (un)informative | −− |
| | Absent | cf > cg > o/rf/rg/eg | eg/o/cg/rg > rf/cf | uninformative | (un)informative | −− |

Table 17: **Comparison of performance-separability classification with and without GPS.** For Proteins, NCI1, and MUTAG, we compare performance separability results with and without including the GPS architecture. The evaluation is based on the performance-separability schema outlined in Table 1.

the self-complementarity of both modes, thus providing insights into the diversity of either the graph structure or the features contained in a given dataset. In particular, extreme $\gamma_{e*}^{1,1}$ scores imply that the metric space of the dual mode is uninteresting (i.e., it lacks any geometric diversity in the distance matrix). We see this phenomenon, for example, in datasets with high $\gamma_{ef}^{1,1}$ such as IMDB-B and IMDB-M, both notably ego-networks. Similarly, we note datasets with extreme $\gamma_{eg}^{1,1}$, including DD, whose node features are derived artificially from node degree. We formalize this notion

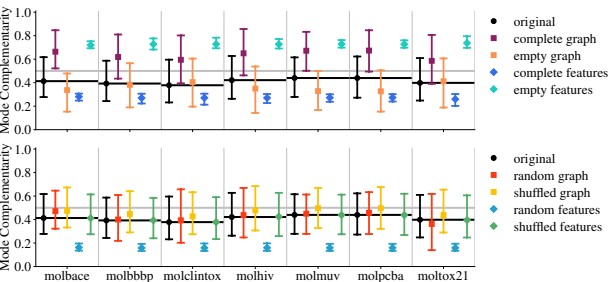

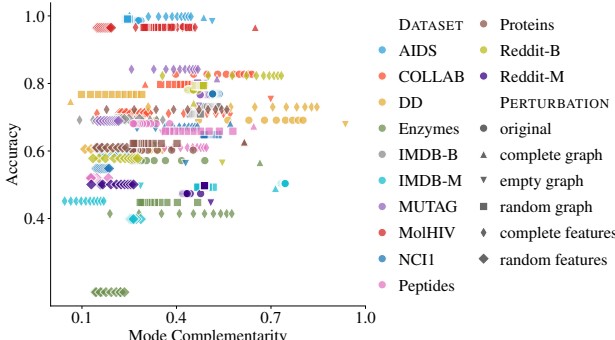

Figure 9: **Comparing *levels of mode complementarity* across different versions of the same dataset.** We show the mean (dot) and 95th percentile intervals (bars) of complementarity scores for the original version as well as 4 deterministic perturbations (top) and 4 randomized perturbations (bottom) of 7 OGB datasets, computed with $t = 1$ diffusion steps. Black horizontal lines indicate mean mode complementarities of the original dataset, and the silver horizontal line marks the $0.5$ threshold relevant for assessing mode diversity. Note that $\gamma_{\text{eg}} = 1 - \gamma_{\text{cg}}$ and $\gamma_{\text{ef}} = 1 - \gamma_{\text{cf}}$ by definition. Contrasting with the variation apparent in Figure 4, the OGB mol-∗ datasets are remarkably similar in their mode-complementarity profiles.

Figure 10: **Mode complementarity and performance: Accuracy as a performance measure.** We show the mean accuracy (y) as a function of the mean mode complementarity (x), for the original version and 5 perturbations of 13 graph-learning datasets, based on our best-on-average models (as in Figure 3). Each marker represents a (dataset, perturbation, $t$) tuple, where $t \in [10]$ is the number of diffusion steps used in the computation of diffusion distances on the graph. Higher mean mode complementarity appears to be associated with higher accuracy, and datasets differ in the range of their mode-complementarity shifts.

of mode diversity in the following section.

### D.3.1. MODE COMPLEMENTARITY AND MODE DIVERSITY

In Table 2, we compute $\Delta^{p,q}_*$ for each graph in the dataset and then compute the mean $\mu$ and the standard deviation $\sigma$ of these scores. A low mean for a mode is indicative of low diversity across the dataset, meaning we see little variation in the geometric structure of the corresponding metric space. To arrive at our symbolic scoring for $\mu$, we divide the interval $[0, 1]$ into five equal-width bins as $[0, 0.2)$, $[0.2, 0.4)$, $[0.4, 0.6)$, $[0.6, 0.8)$, $[0.8, 1.0]$. We proceed similarly for $\sigma$, using smaller brackets due to its different scale: $[0.0, 0.05)$, $[0.05, 0.1)$, $[0.1, 0.15)$, $[0.15, 0.2)$, $[0.2, 1.0)$. As with our symbolic performance-separability scoring, these categories primarily serve to convey our main message, but the numerical values (which do not exhibit discontinuities) should be preferred when conducting in-depth dataset evaluations.

Simply put, datasets with (very) good diversity are candidates for new tasks that could leverage the diversity seen in these modes. In particular, we are interested in datasets that performed poorly in the performance separability evaluation (see Table 1) but have high structural diversity. This type of dataset may have potential as a graph-learning benchmark—if remodeled or assigned a new task that better leverages the information in both modes. As depicted in the second row of our taxonomy table in Section 4.3, these datasets are AIDS, DD, MUTAG, Reddit-B, and Reddit-M.

### D.3.2. MODE COMPLEMENTARITY AND PERFORMANCE

Our intuition is that datasets with high complementarity (different and high information content across the two modes) should perform better. In line with this expectation, we observe a generally positive relationship between mode complementarity and performance when measured by both AUROC and accuracy, as visualized in Figure 5 and Figure 10, respectively. This correlation is further quantified in Table 18, where we symbolically categorize the observed correlations by creating five equal-width bins in the interval $[-1.0, 1.0]$, i.e., $[-1.0, -0.6)$, $[-0.6, -0.2)$, $[-0.2, 0.2)$, $[0.2, 0.6)$, $[0.6, 1.0]$. Again, this categorization only serves to communicate our main message (here: that the correlations are mostly positive and substantively robust to different ways of measurement), but the numerical values should be consulted for gaining detailed insights.

Taking a closer look Figure 5, we can observe how the relationship between mode complementarity and performance changes across mode perturbations (drawn as different shapes). Note that most datasets categorized as good graph-learning benchmarks (such as Peptides and NCI1) exhibit a stronger positive trend between complementarity and performance among their perturbations. We see an even stronger association with some datasets when the original mode (denoted by a circle) is not only the best performer but also has the highest complementarity.

For a given perturbed dataset, Figure 5 also shows the changes over different diffusion steps (i.e., the horizontal

| | Accuracy Values | | | AUROC Values | | | Accuracy | | | AUROC | | | Evaluation |
|---|---|---|---|---|---|---|---|---|---|---|---|---|---|
| Dataset | $p$ | $s$ | $k$ | $p$ | $s$ | $k$ | $p$ | $s$ | $k$ | $p$ | $s$ | $k$ | |
| AIDS | 0.51 | 0.33 | 0.23 | 0.44 | 0.29 | 0.19 | + | + | + | + | + | ∘ | + |
| COLLAB | 0.21 | 0.28 | 0.19 | 0.22 | 0.25 | 0.18 | + | + | ∘ | + | + | ∘ | + |
| DD | -0.17 | -0.31 | -0.20 | -0.27 | -0.24 | -0.16 | ∘ | − | ∘ | − | − | ∘ | − |
| Enzymes | 0.48 | 0.41 | 0.29 | 0.47 | 0.38 | 0.26 | + | + | + | + | + | + | + |
| IMDB-B | 0.33 | 0.28 | 0.19 | 0.57 | 0.50 | 0.34 | + | + | ∘ | + | + | + | + |
| IMDB-M | 0.50 | 0.59 | 0.40 | 0.58 | 0.67 | 0.46 | + | + | + | + | ++ | + | + |
| MUTAG | 0.32 | 0.29 | 0.20 | 0.11 | 0.12 | 0.08 | + | + | + | ∘ | ∘ | ∘ | + |
| MolHIV | 0.14 | 0.19 | 0.13 | 0.61 | 0.55 | 0.39 | ∘ | ∘ | ∘ | ++ | + | + | + |
| NCI1 | 0.66 | 0.69 | 0.48 | 0.47 | 0.59 | 0.40 | ++ | ++ | + | + | + | + | ++ |
| Peptides | 0.65 | 0.37 | 0.25 | 0.71 | 0.52 | 0.36 | ++ | + | + | ++ | + | + | ++ |
| Proteins | 0.21 | 0.20 | 0.12 | 0.09 | 0.10 | 0.07 | + | ∘ | ∘ | ∘ | ∘ | ∘ | ∘ |
| Reddit-B | 0.44 | 0.32 | 0.24 | 0.12 | 0.30 | 0.21 | + | + | + | ∘ | + | + | + |
| Reddit-M | -0.36 | -0.46 | -0.33 | -0.33 | -0.36 | -0.24 | − | − | − | − | − | − | − |

Table 18: **Mode complementarity and performance: Correlation statistics.** We show the Pearson correlation between mode complementarity and test accuracy resp. AUROC for 13 graph-learning datasets, taken over 5 perturbations and $t \in [10]$ diffusion steps, based on our best-on-average models (as in Figure 3).

movement of points with the same color and shape). This has a variable effect on the perturbed $\gamma_*^{1,1}$ score. This variation merits further investigation, but we can already note some initial interesting patterns. For example, the $\gamma_{\varphi_{\mathrm{rf}}}^{1,1}$ scores for MolHIV, Peptides, ENZYMES, MUTAG hardly change over diffusion, while the $\gamma_{\varphi_{\mathrm{cf}}}^{1,1}$ score is more sensitive to the diffusion process. This may indicate that the metric spaces that arise from the graph modes are more similar to the metric spaces that arise from random features. This would suggest using $\varphi_{\mathrm{cf}}$ (as an "uninteresting" metric comparison) to pick up more signal over a diffusion process that occurs in GNNs.

## E. Related Work

Extending the discussion of related work begun in the main text, there are three relevant related lines of work followed in the graph-learning community, namely, (i) *data-centric and multiverse approaches in machine learning*, (ii) *graph-learning benchmark datasets*, and (iii) *graph-learning evaluations*. Overall, we find that RINGS provides a unique perspective on the challenges discussed in these fields.

**Data-Centric and Multiverse Approaches.** This category comprises works that assume a data-centric perspective, potentially imbued with the *multiverse* notion, i.e., the notion that *any* data-analysis task necessitates an elaborate analysis of choices and "non-choices," thus resulting not in a *single* outcome but a *multiverse* of outcomes. This is a novel perspective, originally arising from psychology, gaining traction in general machine-learning applications (Biderman & Scheirer, 2020; Germani et al., 2023; Simson et al., 2024; Wayland et al., 2024), that serves to highlight the *impact* of different decisions, such as data preprocessing or model

selection, on the outcome. Even more broadly, Mazumder et al. (2023) present a call for *data-centric machine learning*, emphasizing the need for considering foremost the *data*, including its quality and provenance, in the development cycle of machine-learning models.

A crucial aspect of any data-centric approach is the development of suitable measures or metrics for the comparison of graphs or their respective (latent) representations. We find several prior works here (Lin et al., 2023; Tang et al., 2023; Zambon et al., 2020), but by their design, such measures cannot focus *beyond* the comparison of individual graphs, remaining instead *intrinsic* with respect to a specific dataset. Our *mode complementarity* overcomes this restriction by adopting a metric-space perspective. While some aspects of metric spaces based on graph structure have been studied (Chuang & Jegelka, 2022; Sanmartín et al., 2022; Sonthalia & Gilbert, 2020; Taha et al., 2023), the focus lies on *embeddings* or *robust shortest paths*, whereas our work is concerned with harnessing metric-space information to provide insights into the *interplay* of graph structure and node features. Thus, our framework might also benefit from integrating *metric space magnitude* (Limbeck et al., 2024),

**Graph-Learning Benchmark Datasets.** Several publications also present new benchmark datasets, driven either by the observation that existing datasets do not cover a sufficiently "dense" part of the graph-learning landscape (Palowitch et al., 2022), or aiming to present more challenging tasks that serve to explore the limitations of existing architectures (Akbiyik et al., 2023; Dwivedi et al., 2022). For example, in the context of *node-classification tasks*, new datasets are proposed to assess the performance of GNNs in heterophilous (or, more precisely, non-homophilous) regimes (Lim et al., 2021; Luan et al., 2022; 2023; Mao

et al., 2023; Platonov et al., 2023a;b), typically drawing upon graph-level measures from *network science* (Barabási, 2016; Newman, 2018).

However, with *mode complementarity*, we seek a score that (1) treats graph structure and node features as equal, (2) works on graphs without node labels and does not make any assumptions about the spaces arising from edge connectivity and node features, and (3) specifically informs graph-level learning tasks such as *graph classification*. To the best of our knowledge, we are the first to propose a score fulfilling these desiderata.

**Graph-Learning Evaluations.**   Previous work on *evaluating* graph learning can be broadly categorized into papers that, like ours, criticize the status quo in terms of dataset usage, the data quality as such, or specific aspects of a given GNN architecture. For the first category, we find works that criticize a lack of suitable baseline comparison partners (Cai & Wang, 2018), issues with hyperparameter tuning and model selection (Errica et al., 2020; Tönshoff et al., 2023; Zhao et al., 2020), or problems with data–model mismatch (Chen et al., 2020). The second category comprises works that highlight the use of unsuitable datasets (Bechler-Speicher et al., 2024; Li et al., 2024), a problem that is also of relevance to other areas in machine learning (Ding & Li, 2018; Fenza et al., 2021). The third category, dealing with the shortcomings of existing architectures, is of particular relevance to practitioners, since it either inspires new research directions or provides practical guidance concerning which models to use in a specific application.

Among the different shortcomings, issues inherent to the message-passing paradigm are well-studied (Alon & Yahav, 2021; Di Giovanni et al., 2023; Yang et al., 2022), often leading to improved architectures (Chen et al., 2022; Han et al., 2023; Michel et al., 2023), with a recent trend being the development of methods that obviate message passing (Fan et al., 2020). Beyond our brief categorization, we also observe interest in general GNN "explainability" strategies (Agarwal et al., 2023; Bonabi Mobaraki & Khan, 2023; Faber et al., 2021; Li et al., 2022; Rathee et al., 2022; Toyokuni & Yamada, 2023; Wang & Shen, 2023; Xie et al., 2022). However one has to bear in mind that such approaches are often tightly coupled to a *specific* task and a *specific* architecture, which, while valuable, cannot help in overcoming dataset deficiencies.

