# OpenReview forum: "No Metric to Rule Them All: Toward Principled Evaluations of Graph-Learning Datasets"
_ICML.cc/2025/Conference — ICML 2025 poster_

### Official Review · Reviewer_5Mv7 · 2025-03-05

**Overall Recommendation:** 4

**Summary:**

This paper is about evaluating graph datasets with the goal of finding good datasets. They argue that  for good datasets, graph structure and node features should have two properties: (P1) they should be task-relevant; (P2) they should contain complementary information. To test this, they propose creating dataset perturbations and measuring their impact: on model performance (P1); or complimentarity of structure and features (P2). By testing a variety of different datasets, the authors make suggestions for how certain datasets should be used in the future.

## update after rebuttal
See my rebuttal comment: This is a good paper that should be accepted -- no changes to my review.

**Claims And Evidence:**

The claims are well support by evidence. The theoretical claims / assumptions (P1, P2) make intuitive sense. The measurement of these properties is done by an expertly designed experiment.

**Essential References Not Discussed:**

No.

**Experimental Designs Or Analyses:**

Yes, the design of measuring P1 and P2 makes sense.

**Methods And Evaluation Criteria:**

The proposed methods (of measuring the impact of P1 / P2) make sense. In, particular the authors base their claims on statistical significance testing, a practice which more machine learning papers should adopt.

The methods and evluation make sense for the goal. Datasets were chosen carefully to cover different domains. Finally, the way GNN hyperparameters were tuned seems remarkably fair (and compute intensive). I have nothing to criticize of the overall experimental design.

However, I feel like the authors could have also investigated the `ZINC` (12k) dataset. As it is used in almost every GNN expressivity paper. Thus, the authors could strengthen and extend the impact of their work by also including this dataset.

**Other Comments Or Suggestions:**

- Definition 2.4 should define that $\varphi$ is a perturbation. It might be easier for the reader to notate a dataset $\mathcal{D}$ perturbed by $\varphi$ as $\varphi (\mathcal{D})$ instead of $\Phi (\mathcal{D})$.
- Definition 2.9: `lifts that take in either structure-based distances arising from G or feature-based distances arising from X`. To me it seems like you intend to create the metric space only from structure or only from the features. However, this definition allows for a combination of both as well, contradicting the explanation.
- You use euclidean distance on node features as a metric. For categorical features, in my opinion this only makes sense if they are one-hot encoded,  is that the case?
- The justifications in the proof of Theorem 2.15 are excellent as they make it easier for the reader to understand the proof. Unfortunately, they reference line numbers that are not part of the proof. As an alternative you could but the reference number on top of the $=$ signs (e.g. $\stackrel{\text{7}}{=} 1- ||D_{d_F}||_{1,1} $).

**Other Strengths And Weaknesses:**

**Strengths:**
- (S1) Our understanding of graph datasets is severely lacking. This work represents an important first step of doing this.
- (S2) The work is built on simple (and in my opinion correct) assumptions (P1, P2). The proposed methods are simple, elegant and easy to understand.
- (S3) The evaluation procedure is extensive (many datasets), fair (extensive hyperparameter tuning) and is based on statistical significance testing.
- (S4) Paper is well written, clear and has a good layout.

**Weaknesses:**
- (W1): Edge features. This work does not take edge features into account.
- (W2): Code for GNNs. As far as I can tell, the released code does not include the training code for the GNNs? This makes it difficult to evaluate this part of the work. In particular, I am interested whether GNNs where trained without taking edge features into account (as this would best align with the framework).
- (W3): The Recommendations (Dataset Taxonomy). One of the key contribution of this work are its recommendations on how to to move forward with how we use datasets (Section 4.3).  This is in my opinion, the most important, impactful and interesting part of this work. However, Section 4.3 is remarkably small and does not properly explain its recommandations. This is particularly problematic for the case of `realigning` datasets, as it is not clear to me what the authors mean by this (e.g. `better data modeling` is very abstract).
- (W4): ZINC (see Methods And Evaluation Criteria).

**To sum up,** this is a good paper that should be accepted.

**Questions For Authors:**

See previous points.

**Relation To Broader Scientific Literature:**

The relation to existing GNN literature is fairly discussed.

**Theoretical Claims:**

I only skimmed the small proofs in the appendix as they do not have a large impact on this work.

---

> ### Author Rebuttal · Authors · 2025-03-30
>
> Thank you for your perceptive comments and your support of our work. In the following, we address the points raised under “Weaknesses” (W1–W4) and “Other Comments Or Suggestions” (Q1–Q4). The point raised under “Methods and Evaluation Criteria” is addressed with W4.
> - **W1 (Accounting for edge features).**
>   We agree that extending our framework to account for edge features would be valuable and will add this to “Future Work” (after ll. 431–434r) in the updated version of the manuscript.
> - **W2 (GNN training code).**
>   - We did not originally include the GNN training code in the reproducibility materials (mostly because the framework itself is already quite hefty) but have now added it to a separate folder (`gnn-training`) in the anonymously shared repository. We will include this code in our public reproducibility package (instead of the stripped-down `src`).
>   - Consistent with our measurement setup, edge features were not taken into account in our experiments, and we will clarify this in the updated manuscript. Note also that not all datasets have edge features to begin with, not all architectures can take them into account, and the PyG implementations of GAT, GCN, and GIN ignore them by default.
> - **W3 (Recommendations/taxonomy).**
>   In a nutshell, our recommendation is “realign” for datasets that do not exhibit performance separability, yet exhibit favorable mode diversity (derived from mode complementarity). In this scenario, there exists interesting variation in the data that graph-learning models can be challenged to leverage in their predictions. However, the existing relationship between this variation and the prediction target, to the extent that it can be picked up by the architectures examined, does not provide a significant performance advantage over settings in which this relationship is deliberately destroyed.
>   “Realignment”, then, collectively denotes several potential operations, including (1) changing the benchmark setting (e.g., datasets with good feature diversity, such as MUTAG or AIDS, could serve as benchmarks for graph-free ML methods), (2) changing the prediction targets (e.g., using different categories to classify discussion threads in Reddit-B and Reddit-M), (3) amending the graph structure (if the dataset lacks structural diversity; this connects to the discussion on graph rewiring), and (4) amending the features (if the dataset lacks feature diversity, as is the case for DD). We will expand the “Dataset Taxonomy” section to clarify our recommendations and their derivation in the updated manuscript.
> - **W4 (ZINC-12k dataset).**
>   Thank you for this suggestion. Since the task associated with ZINC-12k is graph regression, which is commonly evaluated with MAE (rather than accuracy or AUROC), we will take the opportunity to showcase the ability of RINGS to handle graph-regression tasks and include not only ZINC-12k but also two other graph-regression datasets, QM9 and ZINC-250k, adding a separate one-column plot with the graph-regression results to complement Figure 3 (_experiments in progress_). Note that the ZINC-12k dataset (and also QM9 and ZINC-250k) are largely “saturated” (i.e., progress has stalled at low levels of MAE), and it is known that they do not always necessitate a graph-based model, such that we expect high overall performance levels and low performance separability. Thus, we additionally aim to provide results for more challenging molecular benchmark datasets like MOSES and GuacaMol.
> - **Q1 (Definition 2.4).**
>   We originally adopted the uppercase notation to distinguish individual mode perturbations from perturbations applied to a whole dataset but will change this to lowercase following your suggestion. We will also add “and a mode perturbation $\varphi$" after “attributed graphs” in l. 114r of the revised manuscript.
> - **Q2 (Definition 2.9).**
>   While our experiments focus on perturbations that affect either the graph structure or the node features, we deliberately designed our framework to be more general, accommodating also perturbations that would require joint lifts of the structure and the features (as these might help isolate model contributions or dataset characteristics in future work). We will clarify this in the updated manuscript.
> - **Q3 (Handling of categorical node features).**
>   Correct. To match the setup encountered by most graph-learning models, we use the node-feature encodings provided by PyG, which one-hot-encodes categorical node features by default. However, we also appreciate that the one-hot handling of non-metric attributes in most standard graph-learning frameworks itself may not be ideal, and our framework can easily accommodate other distance metrics.
> - **Q4 (Reference numbers).**
>   Thank you for this suggestion, which we will implement in the updated manuscript.
>
> We hope that our responses addressed your concerns and are happy to answer further questions.

---

> > ### Comment · Reviewer_5Mv7 · 2025-04-03
> >
> > Thank you for addressing my comments and especially for uploading the GNN code. I could indeed verify that edge features were not used. I remain of the opinion that this is a good paper that should be accepted. Congratulations to the authors for this cool work.

---

### Official Review · Reviewer_7Dac · 2025-03-07

**Overall Recommendation:** 3

**Summary:**

This paper focuses on the problem of evaluating benchmark datasets for the task of graph classification. In particular, they consider studying the importance that both structure and node features have on performance. They consider measuring this by perturbing the original graph, both in terms of the structures and features, and measuring both the change in performance and change in graph properties. They do so across a variety of common datasets. They find that for many datasets, perturbing either the features or structure has little effect on both performance and mode complementarity. They argue that this indicates a need for newer datasets where both the features are structures are necessary for optimal performance.

**Claims And Evidence:**

I think the claims made in this paper are well supported by the provided evidence.

**Essential References Not Discussed:**

None.

**Experimental Designs Or Analyses:**

I think the experimental design and analyses are generally good.

However, there are two instances where I think they can be improved:

1. One area I think can be improved is which models were used. Currently, only basic GNNs are used in the study. However, many papers have shown that more expressive models can achieve much better performance that basic GNNs on graph-level tasks. More recently, graph transformers have shown very promising performance. I think incorporating 1 or 2 of these methods, such as [1], would enhance the conclusions made by this study. This is especially true when perturbing the graph structure, as these more recent methods shown an enhanced ability to distinguish more complicated graph structural patterns.

2. It's unclear to me what it means when performance and mode separability aren't aligned. For example, for both Reddit datasets, the performance separability is very low, indicating that they are poor datasets. However you then show that the mode diversity is quite high for both structures and features. The authors argue that such datasets may be "misaligned", however it's unclear what this means. I think the authors need to answer: (a) What is meant by misaligned? (b) How can we better align such datasets? (c) If the mode diversity is high, then why is the performance separability so low? I'd argue that this last question is most important as to me it suggests that either: (a) the current methods for measuring $D_F$ and $D_S$ may be lacking (b) Or that the disparity may be due to the methods used. However, if the authors have another idea for the cause I'd be interested.

[1] Rampášek, Ladislav, et al. "Recipe for a general, powerful, scalable graph transformer." Advances in Neural Information Processing Systems 35 (2022): 14501-14515.

**Methods And Evaluation Criteria:**

I think all the proposed method and evaluation are mostly well designed. Specifically,

1. I appreciate that the authors considered multiple types of perturbations. The different levels of perturbation are really helpful for accounting for the various ways that both modifying the features and structure can effect performance.

2. I like that the authors considered a wide variety of datasets, that cover the common ones used for graph classification. This allows for a more comprehensive study. Furthermore, I like the comprehensiveness of the final results.

However, I disagree with how the authors measured performance separability. In Figure 3, the authors compare the final performance by perturbation type. However, I believe that looking at the overall performance is actually not the correct way to check the effect on performance. In reality, for two perturbations, it could be that they actually differ greatly in "how" they achieve similar performance. In an extreme example, let's say we get a 50% accuracy on both the original dataset and one perturbation. Naively, it seems that the perturbation has no effect. However, it could be that they are completely disjoint in which samples they correctly classify.

As such, I think Figure 3 should be supplemented with an additional metric that considers the overlap in correctly classified samples. A simple way would be too calculate the what \% of correct samples for the perturbed dataset were correctly answered by the original dataset . That is, let $S$ be the set of correctly classified samples w/o perturbation. Furthermore, let $S_{\phi}$ be the same under some perturbation $\phi$. We can then calculate $({S_{\phi} \cap S}) / S_{\phi}$, which will be $1$ when all the samples answered correctly by $\phi$ were also correct when using the original dataset. This is of course just one example, the metric itself can vary.

I should be clear, I don't think this will have much of an effect at all on the final results and conclusions made in the paper. However, I believe that the current strategy of just comparing the overall performance is too coarse, and that it's important that we're sure that perturbing the graph either does/doesn't effect the sample-level performance.

**Other Comments Or Suggestions:**

1. I think you can be clearer in the introduction and abstract that you're only focusing on graph classification. This isn't a weakness or anything, just that by reading both the reader may come away with an impression that you study a variety of different graph tasks (as far as I can tell, it is only mentioned at the end of the intro that the focus is graph classification, my apologies if I missed something)

**Other Strengths And Weaknesses:**

None.

**Questions For Authors:**

1. In Figure 3, the accuracy of MolHIV seems to stay unchanged at 100% (or close) across the original dataset and all perturbations. However, the AUC is much lower and varies a lot. I'm curious if the authors have an idea as to what may be causing this discrepancy? Because for all the other datasets, both AUC and accuracy tend to be in lockstep with one another.

**Relation To Broader Scientific Literature:**

I think this paper relates to a lot of work in the area of Graph ML. Particularly, I really like the motivation and the idea behind this study. I think that far too often we neglect the quality of our benchmark datasets, however they are essential for our understanding of the strengths and weaknesses of proposed methods.

**Theoretical Claims:**

Yes, all of them.

---

> ### Author Rebuttal · Authors · 2025-03-30
>
> Thank you for your encouraging feedback. We address your points by the section in which they were raised.
> - **Methods And Evaluation Criteria: Measuring performance separability.**
> We designed performance separability mirroring model-centric evaluation practices to facilitate adoption but agree that a fine-grained perspective will provide additional insights. Following your suggestion, we will include an experiment that examines performance changes at the level of individual graphs in the updated manuscript. We did not log test-set performance at the level of individual graphs in our original experiments but are recomputing the necessary evaluations (*experiments in progress*).
> - **Experimental Designs or Analyses.**
>   - **Coverage of GNN models.**
>   Following your suggestion, we will include results for the GPS and Graphformer models in the updated manuscript (*experiments in progress*).
>   - **Interpreting the discrepancy between performance separability and mode diversity.**
>
>     **(a)** We say that a dataset is “misaligned” if it does not exhibit performance separability, yet exhibits favorable mode diversity (derived from mode complementarity).
>     **(b)** “Realignment” collectively denotes several potential operations, including (1) changing the benchmark setting (e.g., datasets with good feature diversity, such as MUTAG or AIDS, could serve as benchmarks for graph-free ML methods), (2) changing the prediction targets (e.g., using different categories to classify discussion threads in Reddit-B or Reddit-M), (3) amending the graph structure (if the dataset lacks structural diversity; this connects to the graph-rewiring discussion), and (4) amending the features (if the dataset lacks feature diversity, e.g., for DD).
>
>     **(c)** Our experiments show that *higher mode complementarity* is associated with *higher performance*; we do not make any claims about the relationship between *mode diversity* and *performance separability*. Since performance separability assesses the task-specific performance gap between a dataset and its perturbations, and mode diversity measures the task-agnostic variation contained in the modes of an individual dataset, we would not expect high mode diversity to be consistently associated with performance separability (e.g., a dataset could have high structural and feature diversity but low mode complementarity, which could lead to “complete graph” and “complete features” performing on par with the original dataset, eliminating performance separability).
>
>     *Misalignment* occurs when there is interesting variation in the data that models could leverage in their predictions (high mode diversity), but the existing relationship between this variation and the prediction target does not provide a significant performance advantage over settings in which this relationship is deliberately destroyed (lack of performance separability). While the lack of performance separability could also be due to limitations of the models employed (i.e., they may not be expressive enough), given our comprehensive measurement setup, we deem it more likely that the relationship between the variation in the data and the prediction target is strained, prompting the need for *realignment*. However, disentangling in detail the model-related and data-related factors contributing to a lack of performance separability would be a valuable next step to further enhance the RINGS framework.
>
>     We will include these clarifications in the “Dataset Taxonomy” section of our updated manuscript.
>
> - **Other Comments or Suggestions: Clarifying the focus on graph classification.**
> We currently mention the focus on graph classification in ll. 78–80r, ll. 245–247r, ll. 430–433l, and ll. 431–435r. With the inclusion of graph-regression datasets suggested by Reviewer 5Mv7, we will now have two graph-level tasks in our experiments. Hence, we will make the following amendments to clarify our scope:
>   - Abstract ll. 42/43l: Insert “on graph-level tasks” after “extensive set of experiments”.
>   - Introduction l. 80r: Rewrite “extensive experiments on real-world graph-classification datasets” to “extensive experiments, focusing on real-world datasets with graph-level tasks.”
> - **Questions for Authors: Accuracy and AUROC on MolHIV.**
> MolHIV is highly imbalanced, with only 1443 out of 41127 graphs labeled 1 (and the rest labeled 0), leading to consistently high accuracy. AUROC is the officially recommended evaluation metric for this dataset; we include accuracy for completeness only and will note this in the updated manuscript. MolHIV also highlights the need for careful evaluation-metric choices (e.g., given the very skewed class distribution, AUPRC may be preferable over AUROC).
>
> We believe that the improvements to our manuscript prompted by your comments will further strengthen our submission and are happy to address any further questions you may have.

---

> > ### Comment · Reviewer_7Dac · 2025-04-03
> >
> > Thank you. I appreciate the clarifications and the promise of including additional experiments (transformers + individual graph performance).
> >
> > Given that both experiments are still in progress (and they were my main concerns), I will keep my positive score for now. Please let me know if the experiments finish before the end of the rebuttal stage and I will adjust my score accordingly.

---

> > > ### Author Response · Authors · 2025-04-09
> > >
> > > Thank you for confirming your support of our work.
> > > We appreciate your sustained interest in our additional experiments, which we believe will strengthen our evidence base but leave the main message of our paper unaltered.
> > >
> > > While we will happily include our extended results in the updated version of our manuscript, we also value diligence over speed in the design and execution of our experiments.
> > > At this point, we have graph-level performance logs for a subset of our original configurations, and the additional graph-learning models we decided to include following your suggestion are still tuning.
> > >
> > > We share our preliminary results for the graph-level performance comparisons in the newly added `rebuttal` folder in the anonymously shared repository.
> > > In these experiments, for a given dataset, we measure the average similarity between the sets of samples correctly classified by two different (mode perturbation, architecture) configurations A and B, using as our similarity measure either the asymmetric score you proposed (suffix `asymmetric`, dividing by the total number of correct classifications by A) or the Jaccard similarity (suffix `jaccard`, dividing by the cardinality of the union of correct classifications by A and B).
> > >
> > > Our preliminary results include (1) all (mode perturbation, architecture) configurations for MUTAG averaged over 17 seeds, (2) all (mode perturbation, architecture) configurations for Proteins averaged over 15 seeds, and (3) a subset of the (mode perturbation, architecture) configurations for NCI1 averaged over 10 seeds.
> > > For the updated manuscript, we will report statistics for all (mode perturbation, architecture) tuples associated with each dataset, average over 100 random seeds, and include the precise level and variation of our estimates in separate tables (the necessary computations are ongoing).
> > > Using the complete graph-level performance data, we will also be able to identify the individual graphs that are responsible for the performance fluctuations we observe (reducing the "accuracy similarity" defined above), and to examine how the mode complementarity of these graphs impacts their "classifiability" depending on the mode-complementarity distribution of the training (and validation) data.
> > >
> > > Our work aims to, inter alia, raise the standards of experimental hygiene in the graph-learning community.
> > > As we are prudently implementing your suggestions, we hope that our preliminary results convinced you that our original results will hold also in light of our additional experiments, and we are looking forward to including our extended analyses in the updated version of our manuscript.

---

### Official Review · Reviewer_rJcn · 2025-03-14

**Overall Recommendation:** 4

**Summary:**

This paper introduces a novel framework, Rings, for evaluating the quality of graph-learning datasets by quantifying differences between the original dataset and its perturbed representations. The authors propose two key metrics: performance separability and mode complementarity, which assess the relevance and complementary nature of the graph structure and node features for a given task. The framework is applied to 13 popular graph-classification datasets, revealing significant insights into the quality and suitability of these datasets for graph-learning tasks. The paper is well-structured, methodologically sound, and provides actionable recommendations for improving benchmarking practices in graph learning.

**Claims And Evidence:**

The claims are well supported by the empirical studies.

**Essential References Not Discussed:**

N.A.

**Experimental Designs Or Analyses:**

Experiments follow standard procedures in graph learning.

**Methods And Evaluation Criteria:**

The methods and evaluation criteria are appropriate.

**Other Comments Or Suggestions:**

NA.

**Other Strengths And Weaknesses:**

Strengths:

S1. The Rings framework offeres a principled approach to dataset evaluation that goes beyond traditional model-centric evaluations.

S2. The extensive experiments on 13 datasets demonstrate the practical utility of the framework and provide valuable insights into the quality of these graph datasets.

S3. The paper is well-written, with clear explanations of the methodology and rigorous experimental design.


Concerns:

C1. The current framework is primarily focused on graph-level tasks. Extending it to node-level and edge-level tasks could further enhance its applicability.

C2. While the paper provides some theoretical insights, a more comprehensive theoretical analysis of the properties of several key concepts could strengthen the work.

**Questions For Authors:**

In addition to C1 and C2, how does the RING framework affect (promote) the graph model design (under node-/edge-/graph-level tasks)?

**Relation To Broader Scientific Literature:**

This submission can inspire graph learning area, especially its understanding of datasets, which may promote the design of graph models.

**Theoretical Claims:**

N.A.

---

> ### Author Rebuttal · Authors · 2025-03-30
>
> Thank you for your thoughtful comments and your support of our work. In the following, we address C1 and C2 as well as your additional question (“Q1”).
>
> - **C1 (restriction to graph-level tasks).**
> As noted in “Discussion→Future Work” (ll. 431–435r), we see extending our framework to node-level and edge-level tasks as an interesting direction for future work. While assessing performance separability for such tasks will essentially work out-of-the-box, extending the notion of mode complementarity to these settings will require considerable additional research.
> - **C2 (additional theoretical insights).**
> As stated in “Discussion→Future Work” (ll. 427–431r), we agree that further theoretical insights are desirable. However, we estimate that the corresponding analyses will merit a separate paper. We will provide some more guidance on how to arrive at additional theoretical insights (e.g., by taking an information-theoretic perspective) in the “Future Work” paragraph of the updated manuscript.
> - **Q1 (model-design guidance).**
> We believe that RINGS could promote model design in several ways.
>     1. As highlighted by our dataset taxonomy, performance separability can eliminate benchmark datasets that do not require the specific capabilities of graph-learning models to integrate information from both modes, helping the community direct model development to where it is most needed.
>     2. Mode-complementarity and mode-diversity measurements could guide architectural choices for a specific task on a specific dataset (e.g., whether a model should focus on leveraging the graph structure or the node features).
>     3. Mode-complementarity and mode-diversity measurements could inform model designs that incorporate data-centric components to adaptively enhance performance (both at the level of individual observations and at the level of entire datasets)–e.g., by preprocessing the data to increase mode complementarity (which is correlated with performance).
>
>     We will add these suggestions to the updated manuscript but note that 2\. and 3\. merit further experimental confirmation.
>
> We hope that these responses addressed your concerns and are happy to answer any further questions you may have.

---

### Decision · Program_Chairs · 2025-05-01

**Decision:**

Accept (poster)

**Comment:**

This paper introduces a principled and well-motivated framework, RINGS, for evaluating the quality of graph-learning datasets through mode perturbations and dataset ablations. The authors propose novel metrics—performance separability and mode complementarity—to assess dataset utility from both structural and feature perspectives. The paper is clearly written, methodologically rigorous, and backed by extensive experiments on a wide range of benchmarks. The reviewers consistently praise the conceptual novelty, careful experimental design, and actionable insights for dataset curation and model evaluation. While suggestions were made regarding broader task coverage (e.g., node-level tasks, edge features) and additional metrics, the authors provided thoughtful and detailed rebuttals along with plans to incorporate improvements into the final version. Given the significance of the problem and the quality of the contribution, I recommend acceptance.